# PA-GAN: Improving GAN Training by Progressive Augmentation

## Abstract

Despite recent progress, Generative Adversarial Networks (GANs) still suffer from training instability, requiring careful consideration of architecture design choices and hyper-parameter tuning. The reason for this fragile training behaviour is partially due to the discriminator performing well very quickly; its loss converges to zero, providing no reliable backpropagation signal to the generator. In this work we introduce a new technique - *progressive augmentation of GANs (PA-GAN)* - that helps to overcome this fundamental limitation and improve the overall stability of GAN training. The key idea is to gradually increase the task difficulty of the discriminator by progressively augmenting its input space, thus enabling continuous learning of the generator. We show that the proposed progressive augmentation preserves the original GAN objective, does not bias the optimality of the discriminator and encourages the healthy competition between the generator and discriminator, leading to a better-performing generator. We experimentally demonstrate the effectiveness of the proposed approach on multiple benchmarks (MNIST, Fashion-MNIST, CIFAR10, CELEBA) for the image generation task.

## 1 Introduction

Generative Adversarial Networks (GANs) (Goodfellow et al., 2014) are a recent development in the field of deep learning, that have attracted a lot of attention in the research community (Radford et al., 2016; Salimans et al., 2016; Arjovsky et al., 2017; Karras et al., 2018). GANs fall into the category of generative models, i.e. models that allow sampling of new data points from encoded high-dimensional data distributions, such as images. The GAN framework can be formulated as a competing game between the generator and the discriminator. Mathematically, training GANs requires solving a min-max problem. Since both the generator and the discriminator are typically parameterized as deep convolutional neural networks with millions of parameters, optimization is notoriously difficult in practice (Arjovsky et al., 2017; Gulrajani et al., 2017; Miyato et al., 2018).

The difficulty lies in maintaining the healthy competition between the generator and discriminator. A commonly occurring problem arises when the discriminator overshoots, leading to escalated gradients and oscillatory GAN behaviour (Mescheder et al., 2018). As a result the generator fails to learn the multimodal structure of the true distribution. Moreover, the supports of the data and model distributions typically lie on low dimensional manifolds and are often disjoint (Arjovsky & Bottou, 2017). Consequently, there exists a nearly trivial discriminator that can perfectly distinguish real data samples from synthetic ones. Once such a discriminator is produced, its loss quickly converges to zero and the gradients used for updating parameters of the generator become useless.

In this work we introduce a new technique to overcome this problem - *progressive augmentation of GANs (PA-GAN)* - that helps to control the behaviour of the discriminator and thus improve the overall training stability. The key idea is to progressively augment the input of the discriminator network with auxiliary random variables, enlarging the sample space dimensionality, in order to gradually increase the discrimination task difficulty (see Figure 1). In doing so, the discriminator can be prevented from becoming over-confident, enabling continuous learning of the generator. As opposed to standard data augmentation techniques (e.g. rotation, cropping, resizing), the proposed progressive augmentation does not directly modify the data samples, but rather is structurally appended to them. In particular, for the single level augmentation along with the data sample $x$ the discriminator takes also as input the binary random variable $s \in \{0, 1\}$. The class of the augmented sample $(x, s)$ is then set based on the combination $x$ with $s$, resulting in real and synthetic samples contained in both true and fake classes. This presents a more challenging task for the discriminator, as it needs to tell the real and synthetic samples apart and additionally understand the association rule. We can further increase the task difficulty of the discriminator by progressively augmenting its input space and enlarging the dimensionality of $s$ during the course of training.

**Figure 1:** Visualization of progressive augmentation. With each extra augmentation level ($L \to L + 1$) the dimensionality of the discriminator input space is increased and the discrimination task gradually becomes harder. This strategy prevents the discriminator from easily finding a decision boundary between two classes and thus leads to meaningful gradients for the generator updates.

We show that the proposed PA-GAN preserves the original GAN objective and is an outcome of a systematic derivation. In contrast to prior work (Salimans et al., 2016; Sønderby et al., 2017; Arjovsky & Bottou, 2017), it does not bias the optimality of the discriminator (see Sec. 4). Structurally augmenting the input sample space and mapping it to higher dimensions not only challenges the discrimination task, but, in addition, encourages the generator to explore various paths towards the data distribution, leading to improved variation of the generated samples (see Sec. 5).

Our technique is orthogonal to existing work, it can be successfully employed with other regularizations strategies (Miyato et al., 2018; Gulrajani et al., 2017) and different network architectures (Chen et al., 2016; Radford et al., 2016), which we demonstrate in Sec. 5. We experimentally show the effectiveness of the progressive augmentation of GANs for image generation tasks on multiple benchmarks (MNIST (LeCun et al., 1998), Fashion-MNIST (Xiao et al., 2017), CIFAR10 (Krizhevsky, 2009), CELEBA (Liu et al., 2015)) across different evaluation metrics (IS (Theis et al., 2016), FID (Huszár, 2015), KID (Bińkowski et al., 2018)).

## 2    RELATED WORK

Many recent works have focused on improving the stability of GAN training and the overall visual quality of generated samples (Radford et al., 2016; Roth et al., 2017; Karras et al., 2018; Gulrajani et al., 2017; Miyato et al., 2018). As reported by Arjovsky & Bottou (2017), the reason for the unstable behaviour of GANs is partly due to a dimensional mismatch or non-overlapping support between the real data and the generative model distributions, resulting in an almost trivial task for the discriminator. Once the performance of the discriminator is maxed out, it provides a non-informative signal to train the generator. To avoid vanishing gradients, the original GAN paper (Goodfellow et al., 2014) proposed to modify the min-max based GAN objective (MM GAN) to a non-saturating loss (NS GAN). However, even with such a re-formulation the generator updates tend to get worse over the course of training and optimization becomes massively unstable (Arjovsky & Bottou, 2017).

Prior approaches tried to mitigate this issue by using heuristics to weaken the discriminator, such as decreasing its learning rate, adding label or input noise. Salimans et al. (2016) proposed a one-sided label smoothing technique to smoothen the classification boundary of the discriminator, thereby preventing it from being overly confident, but at the same time biasing its optimality. The works of Arjovsky & Bottou (2017) and Sønderby et al. (2017) made the job of the discriminator harder by adding Gaussian noise to both generated and real samples. Moving the manifolds of the data and model distributions closer to each other by adding the input noise ensures a meaningful overlap between their supports, which is desired in order for the generator to eventually approach the data distribution. However, adding high-dimensional noise introduces significant variance in the parameter estimation, which slows down the training and requires multiple samples for counteraction (Roth et al., 2017). Similarly, Sajjadi et al. (2018) proposed to blur the input samples and gradually remove the blurring effect during the course of training. Instead of adding noise to the input, Zhang et al. (2018b) created mixup samples by interpolating between synthetic and real ones, which leads to a more stable behaviour of GANs. These techniques, i.e., additive noise, blurring and sample mixup, perform direct modifications on the data samples.

Another line of work resorts to cost function reformulation to improve the stability of GAN training, e.g. by using the Pearson $\chi^2$ divergence for least square GANs (LS GANs) (Mao et al., 2016), kernel maximum mean discrepancy (MMD) for MMD-GANs (Li et al., 2017; Dziugaite et al., 2015), or f-divergence for f-GANs (Nowozin et al., 2016). Arjovsky et al. (2017) proposed the Wasserstein GAN (WGAN) with the training objective derived from the Wasserstein distance, aiming to

mitigate the vanishing gradient problem. The drawback of this approach is the weight clipping of the discriminator employed to enforce smoothness, which adversely reduces the capacity of the discriminator. Alternative to weight clipping, Gulrajani et al. (2017) added a soft penalty on the gradient norm which ensures a 1-Lipschitz discriminator. The gradient norm penalty can be seen as a weight regularization technique for the discriminator and was shown to improve the performance with other losses as well (Fedus et al., 2018). Similarly, Roth et al. (2017) proposed to add a penalty on the weighted gradient-norm of the discriminator in the context of f-divergences, showing its equivalence to adding input noise. On the downside, regularizing the discriminator with the gradient penalty depends on the model distribution which changes during training and results in increased runtime due to additional gradient norm computation. Miyato et al. (2018) proposed another way to stabilize the discriminator by normalizing its weights and limiting the spectral norm of each layer to constrain the Lipschitz constant. This normalization technique does not require intensive tuning of hyper-parameters and is computationally light. Most recently, Zhang et al. (2018a) showed that spectral normalization is also beneficial for the generator by preventing the escalation of parameter magnitudes and avoiding unusual gradients.

Several methods have proposed to modify the training methodology of GANs in order to further improve stability, e.g. by considering multiple discriminators with different roles (Durugkar et al., 2017) or growing both the generator and discriminator networks progressively (Karras et al., 2018).

In this work we introduce an orthogonal way to stabilize the GAN training by progressively increasing the discrimination task difficulty. To this end, a novel and structured way of augmenting the discriminator input space is proposed. In contrast to other techniques, our method does not bias the optimality of the discriminator or alter the training samples. Furthermore, the proposed augmentation is complementary to prior work. It can be employed with different GAN architectures, adapted to various divergence measures and combined with other regularization techniques (see Sec. 5).

## 3 THEORETICAL BACKGROUND

For generative modeling, one common approach is to adopt divergence measures as loss functions for the generator. Our method belongs to this line of work. In contrast to prior work, our primary focus is not on explicitly minimizing the divergence between the data and model distributions defined on the sample space $\mathcal{X}$. Alternatively, we first structurally augment the training samples (both real and synthetic ones) and then minimize the divergence between distributions defined on the augmented sample space. For computing the divergence, we adopt the adversary process introduced by (Goodfellow et al., 2014) (Sec. 3.1), while the proposed augmentation is inspired by the information theory view of Jensen-Shannon (JS) divergence (Sec. 3.2). Both of them are briefly reviewed in this section to lay the theoretical groundwork for our method, which we then discuss in Sec. 4.

### 3.1 ADVERSARY PROCESS OF GANS

Let $\mathcal{X}$ denote a compact metric space such as the image space $[0, 1]^d$ of dimension $d$. The data distribution $\mathbb{P}_d$ and the model distribution $\mathbb{P}_g$ are both probability measures defined on $\mathcal{X}$. In the context of GANs, $\mathbb{P}_g$ is commonly induced by a function $G$ that maps a random noise vector $\boldsymbol{z}$, following a given prior distribution $\mathbb{P}_z$, to a synthetic data sample, i.e. $\boldsymbol{x}_g = G(\boldsymbol{z}) \in \mathcal{X}$.

The core idea behind GAN training is to set up a competing game between two players, commonly termed the discriminator $D$ and generator $G$. Mathematically, their objective can be formulated as

$$\min_G \max_D L(D, G) \triangleq \mathbb{E}_{\boldsymbol{x} \sim \mathbb{P}_d} \left\{ \log \left[ D(\boldsymbol{x}) \right] \right\} + \mathbb{E}_{\boldsymbol{x} \sim \mathbb{P}_g} \left\{ \log \left[ 1 - D(\boldsymbol{x}) \right] \right\}, \quad (1)$$

with $D : \mathcal{X} \mapsto [0, 1]$. The optimal $D^*$ respectively classifies $\boldsymbol{x} \sim \mathbb{P}_d$ and $\boldsymbol{x} \sim \mathbb{P}_g$ as TRUE and FAKE, i.e. binary classification. Its achieved maximum equals the JS divergence between $\mathbb{P}_d$ and $\mathbb{P}_g$, which is then used as the loss function by the generator to optimize $G$ (Goodfellow et al., 2014).

### 3.2 INFORMATION THEORY VIEWPOINT

Apart from quantifying distributions' similarity, the JS divergence has an information theory interpretation that inspires our approach presented in Sec. 4. In accordance with the binary classification task of the discriminator, we introduce a binary random variable $s$ with a uniform distribution $\mathbb{P}_s$.

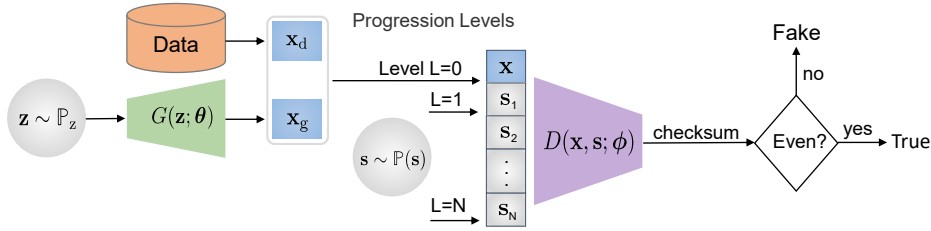

**Figure 2:** Overview of the proposed PA-GANs training. With each level of progressive augmentation $L$ the dimensionality of $s$ is enlarged from 1 to $N$, $s = \{s_1, s_2, \ldots, s_N\}$. The task difficulty of the discriminator, checksum computation of $(x, s)$, increases as the length of $s$ grows.

Associating $s = 0$ and $s = 1$ respectively with $x \sim \mathbb{P}_d$ and $x \sim \mathbb{P}_g$, we obtain a joint distribution

$$\mathbb{P}(x, s) = \mathbb{P}_s(s)\mathbb{P}(x|s) \quad \text{with} \quad \mathbb{P}(x|s) \triangleq \begin{cases} \mathbb{P}_d(x) & \text{if } s = 0 \\ \mathbb{P}_g(x) & \text{if } s = 1 \end{cases} . \tag{2}$$

The marginal distribution with respect to $x$ (a.k.a. the mixture distribution) is equal to

$$\mathbb{P}_m \triangleq \frac{\mathbb{P}_d + \mathbb{P}_g}{2}. \tag{3}$$

Computing the mutual information of the two random variables $s$ and $x$ based on $\mathbb{P}(x, s)$, we have

$$I(x; s) = H(x) - H(x|s) \overset{(a)}{=} 0.5 \int p_d(x) \log p_d(x) \mathrm{d}\mathbb{P}_m(x) + 0.5 \int p_g(x) \log p_g(x) \mathrm{d}\mathbb{P}_m(x)$$

$$= D_{\mathrm{JS}}\left(\mathbb{P}_d \| \mathbb{P}_g\right), \tag{4}$$

where the equality $(a)$ is the outcome of computing the two entropy terms based on the reference measure $\mathbb{P}_m$. The minimum of the JS divergence $D_{\mathrm{JS}}\left(\mathbb{P}_d \| \mathbb{P}_g\right)$ equal to zero is attainable iff $\mathbb{P}_d = \mathbb{P}_g$. This condition makes the joint distribution function $\mathbb{P}(x, s)$ factorizable, indicating the independence between $x$ and $s$, and thereby yielding zero mutual information.

## 4 PROGRESSIVELY AUGMENTED GAN TRAINING

Relying on the information theory view of the JS divergence given in the previous section, we can cast the optimization objective of the generator as mutual information minimization

$$\min_G D_{\mathrm{JS}}\left(\mathbb{P}_d \| \mathbb{P}_g\right) \equiv \min_G I(x; s). \tag{5}$$

Based on (5), in Sec. 4.1 we will first present an equivalent problem to (5), particularly showing how the auxiliary random bit $s$ leads us to a novel and structured way to augment the sample $x \in \mathcal{X}$ for training the discriminator. By further identifying a common principle behind the equivalent problems, in Sec. 4.2 we extend the single level augmentation based on one bit $s$ to progressive multi-level augmentation with an arbitrarily long random bit sequence. Progressively increasing the number of augmentation levels equips us with a new mechanism to balance the two-player competing game. In Sec. 4.3 we describe the integration of the proposed augmentation into neural networks and present how to schedule the augmentation progression during training.

### 4.1 SINGLE LEVEL AUGMENTATION

Starting from the case of single level augmentation with one bit $s$, we first note that the following two minimization problems are equivalent

$$\min_G D_{\mathrm{JS}}\left(\mathbb{P}_d \| \mathbb{P}_g\right) \equiv \min_G D_{\mathrm{JS}}\left(\mathbb{P}(x, s) \| \mathbb{Q}(x, s)\right) \tag{6}$$

where $\mathbb{P}(x, s)$ has been defined in (2) and $\mathbb{Q}(x, s)$ is constructed as

$$\mathbb{Q}(x, s) = \mathbb{P}_s(s)\mathbb{Q}(x|s) \quad \text{with} \quad \mathbb{Q}(x|s) \triangleq \begin{cases} \mathbb{P}_d(x) & \text{if } s = 1 \\ \mathbb{P}_g(x) & \text{if } s = 0 \end{cases} . \tag{7}$$

The two joint distribution functions $\mathbb{P}(x, s)$ and $\mathbb{Q}(x, s)$ differ from each other by the association of the bit $s \in \{0, 1\}$ with the data and synthetic samples, respectively. Their marginals with respect to $x$ are neither the data nor the generative model distribution, but by construction are identical and equal to $\mathbb{P}_m(x)$ as given in (3). It is worth noting that the equivalence holds even if the feasible solution set of $\mathbb{P}_g$ determined by $G$ does not include the data distribution $\mathbb{P}_d$. This is of practical interest as it is often difficult to guarantee the fulfillment of such premise when modelling $G$. For the detailed proof of the equivalence in 6 we refer the reader to App. A.1.

Next, we attempt to solve the equivalent problem given in (6), following the adversary learning process behind GANs. Specifically, the objective of the discriminator $D$ converges to estimation of the JS divergence between $\mathbb{P}(\boldsymbol{x}, s)$ and $\mathbb{Q}(\boldsymbol{x}, s)$

$$\max_D \mathbb{E}_{(\boldsymbol{x}, s) \sim \mathbb{P}} \left\{ \log \left[ D(\boldsymbol{x}, s) \right] \right\} + \mathbb{E}_{(\boldsymbol{x}, s) \sim \mathbb{Q}} \left\{ \log \left[ 1 - D(\boldsymbol{x}, s) \right] \right\}. \tag{8}$$

Comparing with the original discrimination task in GANs, i.e. (1), two differences are worth noting. First, the above discriminator takes $s$ in addition to the sample $\boldsymbol{x} \in \mathcal{X}$ as the input. We, therefore, view $s$ as a single level of augmentation to the sample $\boldsymbol{x}$. Second, the distributions that form the two classes (i.e. TRUE vs. FAKE) become $\mathbb{P}(\boldsymbol{x}, s)$ and $\mathbb{Q}(\boldsymbol{x}, s)$, instead of the original data and model distributions. Based on the definitions in (2) and (7), we identify $(\boldsymbol{x}_{\mathrm{d}}, s = 0)$ and $(\boldsymbol{x}_{\mathrm{g}}, s = 1)$ as belonging to the TRUE class, whereas $(\boldsymbol{x}_{\mathrm{d}}, s = 1)$ and $(\boldsymbol{x}_{\mathrm{g}}, s = 0)$ to the FAKE class. Thus, the real samples are no longer always in the TRUE class, and the synthetic samples are no longer always in the FAKE class. TRUE and FAKE now depend on the combination of $\boldsymbol{x}$ with $s$ (see Figure 1).

Here, we introduce a simple trick to easily detect the class of a given pair. Namely, let the data and synthetic samples respectively convey one bit of information, with $\boldsymbol{x}_{\mathrm{d}}$ encoding bit zero and $\boldsymbol{x}_{\mathrm{g}}$ encoding bit one from now on. Then, the checksum of the pair $(\boldsymbol{x}, s)$ determines the respective class, i.e. checksum zero for TRUE and one for FAKE. [1] The checksum computation poses a more challenging task for the discriminator, as it needs to tell the real and synthetic samples apart and additionally understand the checksum rule. Therefore, such augmentation is usable for preventing early maxing-out of the discriminator. More importantly, it does not compromise the core role of the discriminator in GAN training: informing the generator about the difference between the data and generative model distribution. This statement is confirmed by the equivalence at (6).

## 4.2 PROGRESSIVE MULTI-LEVEL AUGMENTATION

We further extend the single level augmentation with one bit $s$ to multi-level augmentation with an arbitrarily long random bit sequence $\boldsymbol{s}$. Note that the two optimization problems on both sides of (6) rely on the JS divergence to quantify the difference of two distributions. Let us replace the data and generative distributions, i.e. $\mathbb{P}_{\mathrm{d}}$ and $\mathbb{P}_{\mathrm{g}}$, respectively with $\mathbb{P}(\boldsymbol{x}, s)$ and $\mathbb{Q}(\boldsymbol{x}, s)$. Following the same line of argumentation, we can systematically add a new bit. Repeating this procedure $L$ times will give us a bit sequence $\boldsymbol{s}$ with length $L$ plus a series of equivalent problems with the same structure

$$\min_G D_{\mathrm{JS}} \left( \mathbb{P}_{\mathrm{d}} \| \mathbb{P}_{\mathrm{g}} \right) \equiv \min_G D_{\mathrm{JS}} \left( \mathbb{P}(\boldsymbol{x}, s_1) \| \mathbb{Q}(\boldsymbol{x}, s_1) \right) \equiv \min_G D_{\mathrm{JS}} \left( \mathbb{P}(\boldsymbol{x}, s_1, s_2) \| \mathbb{Q}(\boldsymbol{x}, s_1, s_2) \right)$$
$$\cdots \equiv \min_G D_{\mathrm{JS}} \left( \mathbb{P}(\boldsymbol{x}, \boldsymbol{s}) \| \mathbb{Q}(\boldsymbol{x}, \boldsymbol{s}) \right). \tag{9}$$

To estimate the JS divergence in (9), the augmented discriminator as defined in (8) now takes a bit sequence $\boldsymbol{s}$ instead of the single bit $s$, in addition to $\boldsymbol{x}$. The combination of the sample $\boldsymbol{x}$ and $\boldsymbol{s}$ yields a multi-level augmentation. Following the analysis of the single bit case, it is not difficult to notice that the checksum mechanism remains. Namely, the discriminator needs to retrieve the one bit information carried by $\boldsymbol{x}$ and then perform a checksum together with the bit sequence $\boldsymbol{s}$. The task difficulty increases as the length of $\boldsymbol{s}$ grows (see Figure 1). Therefore it makes sense to increase the augmentation level by adding more bits, whenever the discriminator becomes too powerful. More importantly, the consistency of the checksum mechanism across different augmentation levels permits progressive augmentation. The same discriminator can be trained from a lower augmentation level and gradually take more bits into consideration (see Figure 2).

## 4.3 IMPLEMENTATION

**Network architecture.** In this work, we aim to maximally reuse existing neural network architectures tailored for GANs, such as DCGAN with spectral normalization (SN DCGAN) (Miyato et al., 2018) and InfoGAN (Chen et al., 2016). According to the above-introduced approach (and Figure 2), the generator architecture can remain unchanged, while the discriminator network requires an alteration to incorporate the augmentation $\boldsymbol{s}$. To this end, we only modify the input layer of the discriminator network, yielding minimal changes.

First, the bit sequence $\boldsymbol{s}$ is preprocessed into a form compatible with $\boldsymbol{x}$. Consider an image sample $\boldsymbol{x}$ with three coordinates, i.e. height, width and color channel (RGB). Each entry of $\boldsymbol{s}$ creates one extra augmentation channel, whereas the bit value is replicated to match the height and width of $\boldsymbol{x}$.

---

[1] By checksum, we mean to conduct the XOR operation over a bit sequence.

It is worth noting that we let each bit take on values $\{0, 1\}$ as input to the network. This choice of values is mainly due to the progressive augmentation during the course of training. When increasing the augmentation level, the additional bit $0$ does not change the checksum and thus the output of $D$. On the contrary, the additional bit $1$ flips the even(odd) checksum to odd(even). An effective change at the discriminator output is necessary to match the discrimination goal, thereby requiring a non-zero input in the first place. Using $\{0, 1\}$ rather than other pairs of values, e.g. $\{-1, 1\}$, helps the discriminator to timely catch the change when progression takes place. Second, we keep the input layer in the network to process $\boldsymbol{x}$ and copy its configuration for processing the reformed $\boldsymbol{s}$. Its kernel size, stride and padding type remain, but the input channel size is changed to $L$ to process each entry of $\boldsymbol{s}$. When a new augmentation level is reached, one extra input channel is instantiated to process the bit $L + 1$. All the following layers of the discriminator remain unchanged.

**Minibatch discrimination.** The gradients for updating $D$ are computed from the loss function given in (8), where the augmentation bit $s$ is replaced by the bit sequence $\boldsymbol{s}$ of length $L$ depending on the current augmentation level. The two expectations are approximated by using minibatches. Each minibatch is constructed with the same number of real data samples, synthetic samples and bit sequences. Each bit sequence is randomly sampled and associated with one real and one synthetic sample. By computing the checksums of the formed pairs, we can decide the correct class of each pair $(\boldsymbol{x}, \boldsymbol{s})$ in the minibatch and feed it into the discriminator to compute the cross-entropy loss. This way of generating $(\boldsymbol{x}, \boldsymbol{s})$ guarantees a balanced number of TRUE/FAKE samples.

**Non-saturating loss (NS).** When employing non-saturating loss for $G$ in the experiments, we follow the reformulation introduced by (Goodfellow et al., 2014). Since the two expectation terms in (8) depend on $G$, this reformulation is applied for both of them, namely

$$\min_G -\mathbb{E}_{(\boldsymbol{x}, \boldsymbol{s}) \sim \mathbb{P}} \left\{ \log\left[1 - D(\boldsymbol{x}, \boldsymbol{s})\right] \right\} - \mathbb{E}_{(\boldsymbol{x}, \boldsymbol{s}) \sim \mathbb{Q}} \left\{ \log\left[D(\boldsymbol{x}, \boldsymbol{s})\right] \right\}. \tag{10}$$

**Progression scheduling.** Bińkowski et al. (2018) introduced the kernel inception distance (KID) to quantify the quality of the synthetic samples and proposed to reduce the learning rate by tracking the reduction of KID over iterations. Here we use KID to decide if the performance of $G$ at the current augmentation level saturates or even starts degrading (typically happens when $D$ starts overfitting or becomes too powerful). Specifically, after $t$ discriminator iterations [2] , we compute the KID between synthetic samples and data samples drawn from the training set. If the current KID score is less than $5\%$ of the average of the two previous ones attained at the same augmentation level, the augmentation is leveled up, i.e. $L$ is increased by one.

Once reaching a new augmentation level, we introduce the following mechanisms to assist the discriminator in quickly picking up the change in the input space. First, the new augmentation bit is generated from a non-uniform distribution, i.e. $\mathbb{P}(s = 1) = p$ and $\mathbb{P}(s = 0) = 1 - p$ with $p < 0.5$. As mentioned before, critical changes on the discriminator side are required for bit $1$. For it to gradually comprehend the new bit, we on purpose create more 0s than 1s and gradually increase $p$ up to $0.5$ (the uniform case) after a certain number of iterations. A simple linear model is adopted

$$p = \min\{0.5 * (t - t_{\text{st}})/t_{\text{r}}, 0.5\} \tag{11}$$

where $t$ and $t_{\text{st}}$ are the current iteration index and the iteration index when the current augmentation level is started, respectively; and $t_{\text{r}}$ controls the rate of increase. It is important to note that $p \neq 0.5$ does not cause unbalanced TRUE and FAKE classes in the constructed minibatches. It only introduces some bias in the generation of the new augmentation bit.

Finally, it is advisable to slow down the learning rate of $G$ when a new augmentation level is reached. When $D$ is not properly adjusted to the new level, its feedback to $G$ can be noisy. For instance, when using the Adam optimizer (Kingma & Ba, 2015), we reset the time step recorded by the $G$ optimizer.

## 5 EXPERIMENTS

**Datasets.** In our experiments we consider MNIST (LeCun et al., 1998), Fashion-MNIST (Xiao et al., 2017), CIFAR10 (Krizhevsky, 2009) and CELEBA (Liu et al., 2015) datasets, with the training set sizes equal to 60k, 60k, 50k and 162k respectively.

---

[2]Each update of $D$ parameters counts as one discriminator iteration. We assume that the update frequency for $D$ can be as or more frequent than that of $G$, but not the opposite.

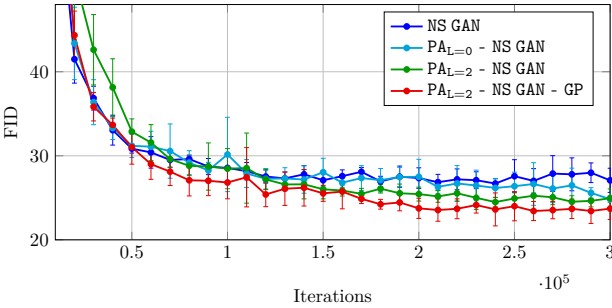

| | 200k | 300k |
|---|---|---|
| NS GAN (Kurach et al., 2018) | 26.7 | — |
| NS GAN (ours) | 26.3 | 25.7 |
| NS GAN - GP (Kurach et al., 2018) | 26.2 | — |
| $PA_{L=0}$ - NS GAN | 25.7 | 24.6 |
| $PA_{L=2}$ - NS GAN | 24.5 | 23.8 |
| $PA_{L=2}$ - NS GAN - GP | **23.2** | **22.5** |

**Figure 3:** Mean FID values attained over iterations with different NS GAN variants across five runs on CIFAR10. The maximal number of achieved PA levels is 7.

**Table 1:** Median FID values attained with different NS GAN variants on CIFAR10. Applying PA and GP on top of NS GAN reduces FID by $\sim 12.5\%$.

**Network architectures.** We employ two well established deep convolutional GAN architectures, SN-DCGAN (Miyato et al., 2018) and InfoGAN (Chen et al., 2016) (see Appendix A.2 for detailed configurations). As they respectively employ spectral normalization (SN) and batch normalization (BN), our aim is to explore the compatibility of PA-GAN with these normalization techniques.

**Evaluation metrics.** We use Fréchet inception distance (FID) (Huszár, 2015) as the primary performance evaluation metric.[3] Additionally, we also report inception score (IS) (Theis et al., 2016) and kernel inception distance (KID) (Bińkowski et al., 2018) in Appendix A.4. For quantifying the quality of synthetic samples, all measures are computed based on 10k test data and 10k synthetic samples, following the evaluation framework of Lučić et al. (2018) and Kurach et al. (2018).[4]

**Training details.** We use the minibatch size of 64 and the Adam optimizer (Kingma & Ba, 2015) with the default setting: $\beta_1 = 0.5$, $\beta_2 = 0.999$ and learning rate $2 \times 10^{-4}$ for both the generator and discriminator, which have an equal update rate. The dimension of $z$ is set to 64 and 128 respectively for InfoGAN and SN-DCGAN. The prior distribution $\mathbb{P}_z$ is uniform. For scheduling progressive augmentation KID is evaluated every $10^4$ discriminator iterations, using 10k generated samples and 10k samples randomly drawn from the training set, and $t_r$ in (11) is set to $5 \times 10^3$.

## 5.1 CIFAR10 WITH SN-DCGAN

In this experiment, we evaluate the progressive augmentation (PA) with NS GAN (GAN with the non-saturating loss) using the SN-DCGAN architecture on CIFAR10. We analyze the benefits of applying PA on top of NS GAN, experiment with starting PA from different augmentation levels and investigate the complementarity of using both PA and the gradient penalty regularization (GP) (Gulrajani et al., 2017). For fair comparison, we follow the experimental setup of (Kurach et al., 2018).

**NS GAN with PA.** Figure 3 and Table 1 compare NS GAN results with and without applying PA. We are able to closely reproduce the NS GAN results reported in (Kurach et al., 2018, Table 6), after 200k iterations we obtain the median FID value of 26.3 vs. original 26.7. By applying PA on top of NS GAN and starting from the augmentation level 0 ($PA_{L=0}$ - NS GAN), we achieve superior performance, with the median FID of 25.7 vs. 26.3 of NS GAN. Training for extra 100k iterations boosts the performance of $PA_{L=0}$ - NS GAN (25.7 vs. 24.6). Starting PA from the level 2 ($PA_{L=2}$ - NS GAN) further improves the median FID (23.8 vs. 25.7); as CIFAR10 contains diverse images a start from a higher augmentation level is recommended. It is worth noting that at early iterations (up to 50k) $PA_{L=2}$ - NS GAN has worse performance than NS GAN. Starting at the augmentation level 2 imposes a more challenging task for the discriminator, thereby showing slower improvement at initial iterations but being beneficial in the longer term. The FID value of $PA_{L=2}$ - NS GAN saturates at a slower pace than NS GAN leading to better overall results.

**NS GAN with PA and GP.** For GP the interpolates are created analogously to (Gulrajani et al., 2017), i.e. $[\tilde{x}, \tilde{s}] = \alpha[x, s]_{\text{TRUE}} + (1 - \alpha)[x, s]_{\text{FAKE}}$ with $\alpha \sim U(0, 1)$. Note that interpolation takes place in the augmented space $[x, s]$, yielding $\tilde{s} \in [0, 1]^L$.

In (Kurach et al., 2018), when GP is applied on top of NS GAN a marginal improvement is observed (26.2 vs. 26.7). However, employing PA results in a more noticeable gain for GP (22.5 vs. 23.8). Furthermore, GP helps to accelerate the learning speed of $PA_{L=2}$ - NS GAN at the initial iterations.

---

[3]How to precisely evaluate the performance of GANs is an open question in itself. From the comparison conducted in (Borji, 2018), FID is considered to be the most informative of the measures.

[4]https://github.com/google/compare_gan

|  | MNIST | Fashion-MNIST | CIFAR10 | CELEBA |
|---|---|---|---|---|
| MM GAN (Goodfellow et al., 2014) | $9.8 \pm 0.9$ | $29.6 \pm 1.6$ | $72.7 \pm 3.6$ | $65.6 \pm 4.2$ |
| LS GAN (Mao et al., 2016) | $7.8 \pm 0.6$ | $30.7 \pm 2.2$ | $87.1 \pm 0.9$ | $53.9 \pm 2.8$ |
| WGAN (Arjovsky et al., 2017) | $6.7 \pm 0.4$ | $21.5 \pm 1.6$ | $55.2 \pm 2.3$ | $41.3 \pm 2.0$ |
| DRAGAN (Kodali et al., 2017) | $7.6 \pm 0.4$ | $27.7 \pm 1.2$ | $69.8 \pm 2.0$ | $42.3 \pm 3.0$ |
| NS GAN (Goodfellow et al., 2014) | $6.8 \pm 0.5$ | $26.5 \pm 1.6$ | $58.5 \pm 1.9$ | $55.0 \pm 3.3$ |
| PA - NS GAN | $8.8 \pm 1.1$ | $18.4 \pm 1.5$ | $44.6 \pm 1.9$ | $46.9 \pm 3.3$ |
| PA - NS GAN (*) | $\mathbf{6.6 \pm 0.8}$ | $\mathbf{15.8 \pm 1.1}$ | $\mathbf{43.1 \pm 1.6}$ | $46.8 \pm 3.2$ |
| WGAN - GP (Gulrajani et al., 2017) | $20.3 \pm 5.0$ | $24.5 \pm 2.1$ | $55.8 \pm 0.9$ | $30.0 \pm 1.0$ |
| PA - WGAN - GP | $13.9 \pm 1.5$ | $26.4 \pm 2.8$ | - | $29.2 \pm 1.7$ |
| PA - WGAN - GP (*) | $8.6 \pm 1.1$ | $20.7 \pm 2.1$ | - | $\mathbf{29.1 \pm 1.7}$ |

**Table 2:** FID values achieved by the listed algorithms with the InfoGAN architecture. The numbers except for PA are taken from (Lučić et al., 2018). All numbers are based on 50 independent runs. Outliers are not removed for PA, see A.7 for further discussion. For the four datasets (from left to right), the results are attained after 20, 20, 100 and 40 epochs, respectively, except for the PA results marked with (*). For (*) the training time is not constrained by the previously specified number of epochs, see A.5 for details.

As indicated in Figure 1, in the augmented space (e.g., $L \in \{1, 2\}$), more paths are created for the generative model to approach the data distribution. GP can help to smoothen the decision boundaries along these paths. As a result, GP and PA jointly improve the performance of NS GAN, being mutually beneficial (22.5 vs. 26.3). In A.3, we additionally report results on CELEBA-HQ (Karras et al., 2018). The enlarged performance gain (18.1 vs. 28.5) reveals a great potential of employing PA for high resolution datasets.

## 5.2 COMPARISONS AMONG DATASETS AND GAN-TYPE ALGORITHMS

Lučić et al. (2018) compared various GAN-type algorithms under the InfoGAN architecture and reported their FID scores after a wide range of hyper-parameter searching. Following their experimental setup, we select hyper-parameters within the candidate set considered by Lučić et al. (2018) (see Appendix A.5 for details) and evaluate PA with NS GAN as well as Wasserstein GAN with GP (WGAN - GP). The maximal number of augmentation levels achieved by PA corresponds to seven. The results are reported in Table 2. Overall, we observe significant gains of employing PA with NS GAN and WGAN - GP across different datasets. Note that the numbers provided by Lučić et al. (2018) are outcomes of removing outliers up to 20% among the fifty independent runs. Since the outlier removing ratio is not specified for individual cases in (Lučić et al., 2018), in Table 2 we report the PA results without removing outliers, see A.7 for further discussion.

As shown in Table 2, the NS GAN performance is quite stable across the datasets. By applying PA with NS GAN, we achieve a pronounced improvement, particularly when the dataset (and hence the image generation task) becomes more complicated. Note that the PA - NS GAN performance on MNIST is worse than NS GAN. This is mainly because NS GAN already performs very well on such a simple dataset and PA requires additional iterations to reach similar results or further improve them as in PA - NS GAN (*). In contrast to NS GAN, WGAN - GP is sensitive to the dataset as shown in Table 2 and reported by Mescheder et al. (2018). However, applying PA helps to stabilize the WGAN - GP performance across different benchmarks. Similarly to NS GAN, longer training leads to better results for PA with WGAN - GP. The results in Table 2 indicate that PA generalizes well across different distance measures and is not limited to the JS divergence.

Besides the experiments in Sec. 5.1 and 5.2, we provide a careful investigation of PA itself in Appendix. Ablation studies on the effect of the progression scheduling and the linear model 11 are given in A.6, followed by the analysis of PA training stability in A.7. The increase of the task difficulty of the discriminator with PA is examined in A.8. In addition to GP, we also compare PA with the dropout regularization in A.9.

## 6 CONCLUSION

In this work we have proposed a novel method - progressive augmentation (PA) - to improve the stability of GAN training, and showed a way to integrate it into existing GAN architectures with minimal changes. Different to standard data augmentation our approach does not modify the training samples, instead it progressively increases the dimension of the discriminator input space by

augmenting it with auxiliary random variables. Higher sample space dimensionality helps to entangle the discriminator and thus to avoid its early performance saturation. Moreover, in the augmented space the generator can explore more paths to approach the data distribution, improving variation of the generated samples. We experimentally have shown pronounced performance improvements of employing the proposed PA with state-of-the-art GAN methods across multiple benchmarks. We demonstrated that PA generalizes well across different network architectures and loss functions and is complementary to other regularization techniques. For future work, we find a joint optimization of PA with neural architectures an interesting direction, for instance, combining it with progressive growing of GANs (Karras et al., 2018). Apart from generative modeling, our approach can also be exploited for semi-supervised learning, generative latent modeling and transfer learning.

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

## A    APPENDIX

### A.1    PROOF OF THE EQUIVALENCE IN (6)

In Sec. 3.2, we have provided the connection of the JS divergence to the mutual information between the auxiliary random bit $s$ and the sample $\boldsymbol{x}$ following the joint distribution $\mathbb{P}(\boldsymbol{x}, s)$ as given in (2)

$$D_{\text{JS}}\left(\mathbb{P}_{\text{d}}\|\mathbb{P}_{\text{g}}\right) = \text{I}(\boldsymbol{x}; s). \tag{12}$$

Interchanging the position of $\mathbb{P}_{\text{d}}$ and $\mathbb{P}_{\text{g}}$ in constructing $\mathbb{P}(\boldsymbol{x}, s)$, we obtain the joint distribution $\mathbb{Q}(\boldsymbol{x}; s)$ as given in (7). With respect to $\mathbb{Q}(\boldsymbol{x}; s)$, we further compute and denote the mutual information between $\boldsymbol{x}$ and $s$ as $\tilde{\text{I}}(\boldsymbol{x}; s)$. Analogous to 4 in Sec. 3.2, we can show

$$D_{\text{JS}}\left(\mathbb{P}_{\text{d}}\|\mathbb{P}_{\text{g}}\right) = \tilde{\text{I}}(\boldsymbol{x}; s). \tag{13}$$

The equality (12) and (13) jointly yield

$$D_{\text{JS}}\left(\mathbb{P}_{\text{d}}\|\mathbb{P}_{\text{g}}\right) = \frac{1}{2}\text{I}(\boldsymbol{x}; s) + \frac{1}{2}\tilde{\text{I}}(\boldsymbol{x}; s) \tag{14}$$

followed by rewriting mutual information as KL divergence

$$D_{\text{JS}}\left(\mathbb{P}_{\text{d}}\|\mathbb{P}_{\text{g}}\right) = \frac{1}{2}D_{\text{KL}}\left(\mathbb{P}(\boldsymbol{x}, s)\|\mathbb{P}_{\text{m}}(\boldsymbol{x})\mathbb{P}_{\text{s}}(s)\right) + \frac{1}{2}D_{\text{KL}}\left(\mathbb{Q}(\boldsymbol{x}, s)\|\mathbb{P}_{\text{m}}(\boldsymbol{x})\mathbb{P}_{\text{s}}(s)\right). \tag{15}$$

It is noted that the marginals of $\mathbb{P}(\boldsymbol{x}, s)$ and $\mathbb{Q}(\boldsymbol{x}, s)$ with respect to $\boldsymbol{x}$ are identical and equal to $\mathbb{P}_{\text{m}}(\boldsymbol{x})$ as given in (3), whereas $\mathbb{P}_{\text{s}}(s)$ is their marginal with respect to $s$.

With the identification of

$$\mathbb{P}_{\text{m}}(\boldsymbol{x})\mathbb{P}_{\text{s}}(s) = \frac{\mathbb{P}(\boldsymbol{x}, s) + \mathbb{Q}(\boldsymbol{x}, s)}{2} \tag{16}$$

we reach to

$$D_{\text{JS}}\left(\mathbb{P}_{\text{d}}\|\mathbb{P}_{\text{g}}\right) = \frac{1}{2}D_{\text{KL}}\left(\mathbb{P}(\boldsymbol{x}, s)\|0.5\mathbb{P}(\boldsymbol{x}, s) + 0.5\mathbb{Q}(\boldsymbol{x}, s)\right)$$
$$+ \frac{1}{2}D_{\text{KL}}\left(\mathbb{Q}(\boldsymbol{x}, s)\|0.5\mathbb{P}(\boldsymbol{x}, s) + 0.5\mathbb{Q}(\boldsymbol{x}, s)\right). \tag{17}$$

The right-hand side term of the above equality is simply the JS divergence between $\mathbb{P}(\boldsymbol{x}, s)$ and $\mathbb{Q}(\boldsymbol{x}, s)$. Since the two JS divergences are completely identical, we can use them interchangeably as the objective function while optimizing $\mathbb{P}_{\text{g}}$. With that we conclude the equivalence proof for the two optimization problems in (6).

### A.2    NEURAL NETWORK ARCHITECTURES

Following Kurach et al. (2018) for SN-DCGAN we employed the same architecture as in (Miyato et al., 2018), which we present in Table A1. For the InfoGAN architecture we followed Lučić et al. (2018) and used the network structure of (Chen et al., 2016), which is described in Table A2. For both experiments with SN-DCGAN and InfoGAN we exploited the implementation provided in `https://github.com/google/compare_gan`.

**Table A1:** SN-DCGAN architecture.

**(a)** Discriminator

| Layer | Kernel | Output shape |
|---|---|---|
| Conv, lReLU | $[3, 3, 1]$ | $h \times w \times 64$ |
| Conv, lReLU | $[4, 4, 2]$ | $\frac{h}{2} \times \frac{w}{2} \times 128$ |
| Conv, lReLU | $[3, 3, 1]$ | $\frac{h}{2} \times \frac{w}{2} \times 128$ |
| Conv, lReLU | $[4, 4, 2]$ | $\frac{h}{4} \times \frac{w}{4} \times 256$ |
| Conv, lReLU | $[3, 3, 1]$ | $\frac{h}{4} \times \frac{w}{4} \times 256$ |
| Conv, lReLU | $[4, 4, 2]$ | $\frac{h}{8} \times \frac{w}{8} \times 512$ |
| Conv, lReLU | $[3, 3, 1]$ | $\frac{h}{8} \times \frac{w}{8} \times 512$ |
| Linear | $-$ | $1$ |

**(b)** Generator

| Layer | Kernel | Output shape |
|---|---|---|
| $\boldsymbol{z}$ | $-$ | $128$ |
| Linear, BN, ReLU | $-$ | $\frac{h}{8} \times \frac{w}{8} \times 512$ |
| DeConv, BN, ReLU | $[4, 4, 2]$ | $\frac{h}{4} \times \frac{w}{4} \times 256$ |
| DeConv, BN, ReLU | $[4, 4, 2]$ | $\frac{h}{2} \times \frac{w}{2} \times 128$ |
| DeConv, BN, ReLU | $[4, 4, 2]$ | $h \times w \times 64$ |
| DeConv, Tanh | $[3, 3, 1]$ | $h \times w \times 3$ |

**Table A2:** InfoGAN architecture.

**(a)** Discriminator

| Configuration per Layer |
| --- |
| $4 \times 4$ conv. 64 lReLU, stride 2 |
| $4 \times 4$ conv. 128 lReLU, stride 2, BN |
| Linear, 1024 lReLU, BN |
| Linear, 1 output |

**(b)** Generator

| Configuration per Layer |
| --- |
| Linear, 1024 ReLU, BN |
| Linear, $7 \times 7 \times 128$ ReLU, BN |
| $4 \times 4$ DeConv, 64 ReLu, stride 2, BN |
| $4 \times 4$ Deconv, 1 or 3 channels |

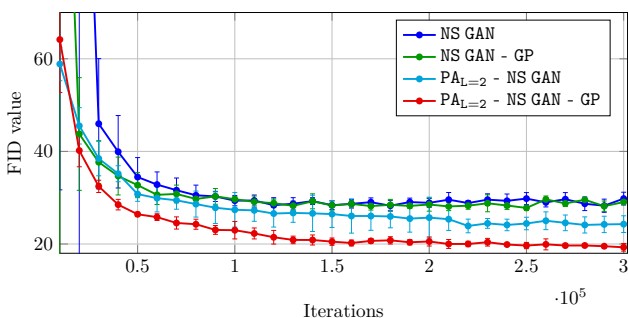

**Figure A1:** Mean FID values attained over iterations across five independent runs with different `NS GAN` variants on CELEBA-HQ ($128 \times 128$), using the SN-DCGAN architecture. The maximal number of achieved augmentation levels is 10.

| | 200k | | 300k | |
| --- | --- | --- | --- | --- |
| | Median | Best | Median | Best |
| `NS GAN` (Kurach et al., 2018) | 31.1 | 29.1 | — | — |
| `NS GAN` (ours) | 27.4 | 26.4 | 27.4 | 26.2 |
| `NS GAN - GP` (ours) | 27.5 | 26.8 | 27.2 | 26.8 |
| $PA_{L=2}$ - `NS GAN` | 23.9 | 22.2 | 23.1 | 21.8 |
| $PA_{L=2}$ - `NS GAN - GP` | **19.7** | **19.4** | **18.8** | **18.1** |

**Table A3:** Median and best FID values attained with different `NS GAN` variants on CELEBA-HQ ($128 \times 128$), using the SN-DCGAN architecture. Applying PA and GP on top of `NS GAN` reduces FID by $\sim 32\%$.

### A.3 CELEBA-HQ WITH SN-DCGAN

Figure A1 and Table A3 report additional results (FID scores) on CELEBA-HQ ($128 \times 128$) (Karras et al., 2018) using the SN-DCGAN network architecture, following the same experimental setup as in Sec. 5.1. The results on CELEBA-HQ ($128 \times 128$) are consistent with our observations in Sec. 5.1. By applying PA on top of `NS GAN - GP` and starting from the augmentation level 2 ($PA_{L=2}$ - `NS GAN - GP`), we achieve superior performance, with the median FID of 18.8 vs. 27.3 of `NS GAN - GP`. Without PA, `NS GAN` and `NS GAN - GP` have almost no FID reduction (or very minor) in the last 100k iterations, whereas PA enables further improvement. The achieved best FID value by $PA_{L=2}$ - `NS GAN - GP` is even $36\%$ smaller than the best FID score achieved by `NS GAN` with ResNet19 reported by (Kurach et al., 2018) (18.1 vs. 28.5).

### A.4 CIFAR10 WITH SN-DCGAN

Figure A2 and A3 together with Table A4 and Table A5 report the inception scores and KID values that are attained following the same experimental setup as in Sec. 5.1 on CIFAR10. The results on both measures are consistent with our observation drawn from the FID values. In the recent work (Arbel et al., 2018), the reported best KID value for CIFAR10 with SN-DCGAN is 0.015, whereas $PA_{L=2}$ - `NS GAN - GP` reduces it to 0.013.

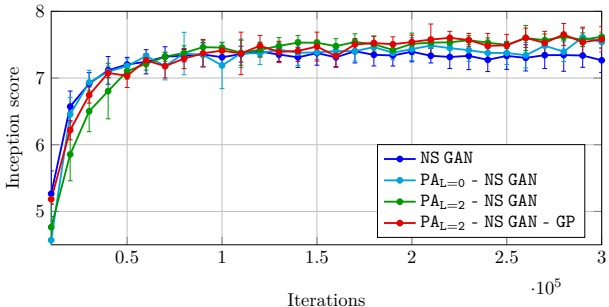

**Figure A2:** Inception scores (IS) attained over iterations with different NS GAN variants averaged across five independent runs on CIFAR10.

|                          | 200k | 300k |
|--------------------------|------|------|
| NS GAN (ours)            | 7.54 | 7.58 |
| PA$_{L=0}$ - NS GAN      | 7.63 | 7.64 |
| PA$_{L=2}$ - NS GAN      | 7.64 | 7.72 |
| PA$_{L=2}$ - NS GAN - GP | 7.59 | 7.76 |

**Table A4:** Median inception scores (IS) attained with different NS GAN variants on CIFAR10.

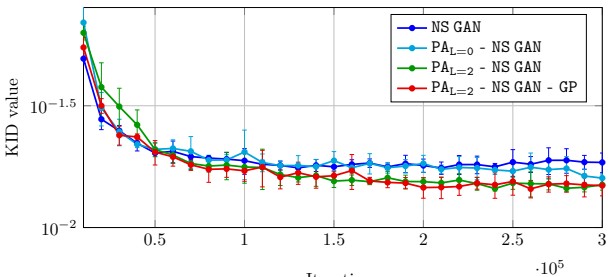

**Figure A3:** KID values attained over iterations with different NS GAN variants averaged across five independent runs on CIFAR10.

|                          | 200k   | 300k   |
|--------------------------|--------|--------|
| NS GAN (ours)            | 0.0168 | 0.0164 |
| PA$_{L=0}$ - NS GAN      | 0.0165 | 0.0153 |
| PA$_{L=2}$ - NS GAN      | 0.0149 | 0.0139 |
| PA$_{L=2}$ - NS GAN - GP | 0.0140 | 0.0133 |

**Table A5:** Median KID values attained with different NS GAN variants on CIFAR10.

Next we present some further remarks on the implementation side. First, to explicitly investigate the combination of GP and PA, we add the GP as an extra regularizer to the discriminator loss function only when the augmentation takes place, and GP is computed with respect to the augmented sample after interpolation, i.e. $[\tilde{x}, \tilde{s}]$. Figures A4 - A6 together with Tables A6 - A8 report results evaluated with the FID, IS and KID metrics. With the use of GP, the starting level of the augmentation has negligible influence on the performance after a sufficient number of iterations.

Second, the chosen hyper-parameters are not optimized for PA. In fact, they are dedicatedly chosen by (Kurach et al., 2018) for NS GAN, i.e. $\beta_1 = 0.5$, $\beta_2 = 0.999$, $\lambda = 1$ (weighting factor for GP) and the learning rate $2 \times 10^{-4}$. We adopt them for PA for the purpose of fair comparison. Therefore, further optimization on the hyper-parameters for PA may potentially yield better results than the reported ones.

Third, in our implementation all of progressively added augmentation channels take on the values $\{0, 1\}$ to ease the progression, as described in Sec. 4.3. For those augmentation channels that are present from the start of training and if the pixel values of the image are normalized to $[-1, 1]$, we accordingly experimented with the values $\{\pm 1\}$. As a result, in the current experiment (i.e., CIFAR10 with SN-DCGAN), we used $\{\pm 1\}$ and switch to $\{0, 1\}$ for subsequent InfoGAN experiment. In general, it is not a critical choice to the performance from our observation.

At last, a set of synthetic images generated by PA$_{L=2}$ - NS GAN - GP with the FID value of 22.5 on CIFAR10 is shown in Figure A7.

## A.5 COMPARISONS AMONG DATASETS AND GAN-TYPE ALGORITHMS

In Sec. 5.2, we apply PA with NS GAN and WGAN - GP and report the achieved FID values in Table 2.

For PA - NS GAN and PA - NS GAN (*) across all datasets we start augmentation with level two, i.e. $L = 2$. The other adopted hyper-parameters for PA - NS GAN are listed in Table A9. It is important to note that they are selected considering the limit on the training epochs specified by Lučić et al. (2018). Allowing more iterations, the performance can be further improved. For instance,

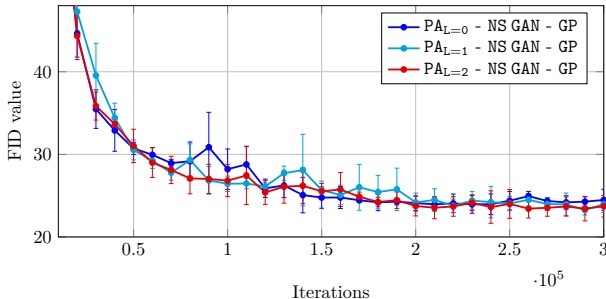

**Figure A4:** FID attained over iterations with different progression starting levels averaged across five independent runs on CIFAR10.

| | 200k | 300k |
|---|---|---|
| NS GAN (ours) | 26.3 | 25.7 |
| PA$_{L=0}$ - NS GAN - GP | 23.2 | 22.8 |
| PA$_{L=1}$ - NS GAN - GP | 23.9 | 23.0 |
| PA$_{L=2}$ - NS GAN - GP | 23.2 | 22.5 |

**Table A6:** FID values attained with different progression starting levels on CIFAR10.

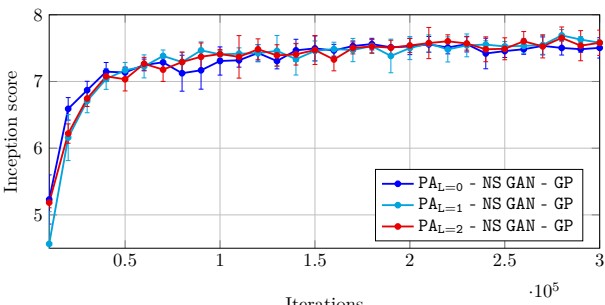

**Figure A5:** IS attained over iterations with different progression starting levels averaged across five independent runs on CIFAR10.

| | 200k | 300k |
|---|---|---|
| NS GAN (ours) | 7.54 | 7.58 |
| PA$_{L=0}$ - NS GAN - GP | 7.63 | 7.67 |
| PA$_{L=1}$ - NS GAN - GP | 7.64 | 7.77 |
| PA$_{L=2}$ - NS GAN - GP | 7.59 | 7.76 |

**Table A7:** Median IS attained with different progression starting levels on CIFAR10.

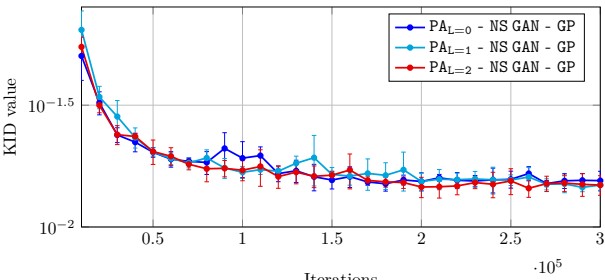

**Figure A6:** KID values attained over iterations with different progression starting levels averaged across five independent runs on CIFAR10.

| | 200k | 300k |
|---|---|---|
| NS GAN (ours) | 0.0168 | 0.0164 |
| PA$_{L=0}$ - NS GAN - GP | 0.0141 | 0.0141 |
| PA$_{L=1}$ - NS GAN - GP | 0.0144 | 0.0141 |
| PA$_{L=2}$ - NS GAN - GP | 0.0140 | 0.0133 |

**Table A8:** Median KID values attained with different progression starting levels on CIFAR10.

PA - NS GAN (*) in Table 2 shows the achieved performance if we run 40k iterations for MNIST and Fashion-MNIST, and run 140k iterations for CIFAR10 and CELEBA. The gains can be further enlarged if the hyper-parameters are adjusted towards the longer training time. For instance, by reducing the learning rate from $10^{-3}$ to $2 \times 10^{-4}$ and performing 140k iterations for MNIST, we can achieve the FID of $4.5 \pm 0.35$.

Proceeding to the case PA - WGAN - GP, the hyper-parameters are listed in Table A10, the batch normalization for the discriminator is disabled. For both MNIST and Fashion-MNIST, the augmentation level starts from one to ensure that it is in place within the 20 epoch training time. For CELEBA, we start from the augmentation level zero and will reach the level one augmentation within the 40 training epochs. Analogous to the previous case, better performance is achievable by PA with the number of iterations increased beyond the original limit, i.e. 40k for MNIST and Fashion-MNIST and 140k for CELEBA.

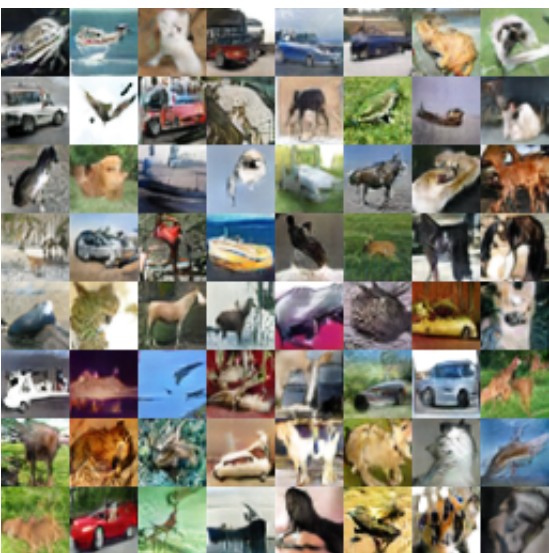

**Figure A7:** Synthetic images generated by $\text{PA}_{\text{L}=2}$ - `NS GAN` - `GP` with the FID 22.5 on CIFAR10.

Finally, we note that to stabilize progression in `WGAN` - `GP` the weighting factor $\lambda$ for GP requires a careful adjustment. In this experiment, we relax $\lambda$ following the idea behind (11), namely by linearly increasing $\lambda$ from zero to its original value within $t_{\text{r}}$ iterations (in this case $t_{\text{r}} = 5 \times 10^3$). Interestingly, we did not find such adaptation necessary when using GP with `NS GAN`. One possible reason is that GP is critical to `WGAN` due to the Lipschitz constraint, but optional to `NS GAN`. Augmentation changes the input space of the discriminator, and the number of terms involved in the GP also increases with the number of augmentation channels. Further investigation on the weighting factor adjustment to fully exploit the benefit of combining PA with `WGAN` - `GP` is a part of our future work. The results reported in this work have confirmed that PA is not limited to the JS divergence.

|  | MNIST | Fashion-MNIST | CIFAR10 | CELEBA |
|---|---|---|---|---|
| $\beta_1$ | 0.5 | 0 | 0.5 | 0.5 |
| $\beta_2$ | 0.999 | 0.999 | 0.999 | 0.999 |
| Learning rate | $10^{-3}$ | $4 \times 10^{-4}$ | $4 \times 10^{-4}$ | $2 \times 10^{-4}$ |

**Table A9:** Hyper-parameters for generating our numbers associated to `PA` - `NS GAN` in Table 2.

|  | MNIST | Fashion-MNIST | CIFAR10 | CELEBA |
|---|---|---|---|---|
| $\beta_1$ | 0 | 0 | - | 0 |
| $\beta_2$ | 0.999 | 0.999 | - | 0.999 |
| $\lambda$ | 0.1 | 0.1 | - | 0.1 |
| Learning rate | $10^{-3}$ | $10^{-3}$ | - | $4 \times 10^{-4}$ |

**Table A10:** Hyper-parameters for generating our numbers associated to `PA` - `WGAN` - `GP` in Table 2.

## A.6 ABLATION STUDIES

**Effect of the Progression Scheduling.** Here we experiment with InfoGAN on CELEBA, as one of the most challenging cases considered in Table 2, to study the effect of the progression scheduling described in Sec. 4.2 and 4.3. Originally, Lučić et al. (2018) suggested 40 epoches (approximately 100k iterations) for the `NS GAN` training. As we can observe from Figure A8, further training can only degrade the performance. One important reason for such behaviour is that the discriminator architecture design of InfoGAN (see Table A2) might be suboptimal for CELEBA (e.g., a fully

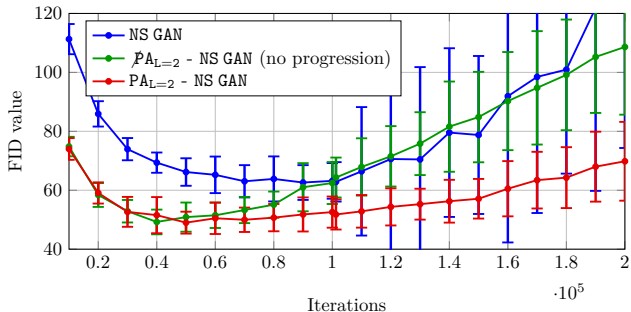

**Figure A8:** FID values with InfoGAN on CELEBA, the results are attained after 40 epochs (about 100k iterations). We remove outliers that are outside 4 standard deviation. In all cases, the number of outliers is less than 5% among 50 independent runs. The maximal number of achieved PA levels is 7.

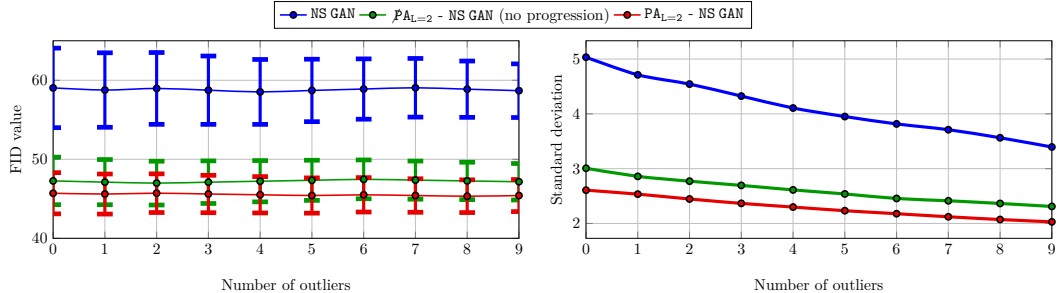

**Figure A9:** FID mean and standard deviation values across 50 independent runs with InfoGAN on CELEBA, dependent on the number of removed outliers.

connected rather than a convolutional layer at the end plus batch normalization conducting minibatch discrimination).[5] By applying the proposed augmentation strategy, even with one augmentation level and no progression, we can achieve much better FID values at much earlier iterations. Moreover, with the progression scheduling of augmentation (from level 2 to 7) we can dramatically slow down the degradation process for this challenging case. We additionally observe that the standard deviation of FID values across 50 runs is much smaller with PA than without it and the gain of the PA remains consistent, independent of the number of outliers removed, see Figure A9.[6] This showcases the positive effect of the proposed PA on the training stability of GANs. Nevertheless, it remains difficult to fully counteract the degradation process caused by the suboptimal network architecture design for the task at hand.

**Role of the Linear Model (11) in PA.** Here we analyse the role of the linear model 11 in progression scheduling described in Sec. 4.3. Figure A10 depicts `D Loss` and `G Loss` over training iterations on CIFAR10 using SN-DCGAN.

Performing PA with a newly augmented bit may lead to the confusion of the discriminator, as EVEN inputs can immediately become ODD ones and vice versa. Empirically, we observe that the loss of the discriminator (i.e., `D Loss`) may stuck at values close to 1.38 for a considerable number of iterations, because the discriminator is confused about the abrupt change. This is particularly true for regularized networks. Whenever such situation takes place, it is harmful for the generator as well. A confused discriminator can no longer guide it towards the data distribution.

Figure A10 depicts the above mentioned confusions after a new augmentation level is introduced. In this case, there are two plateaus of `D Loss` for brown curves which do not employ the linear model 11 in PA, occurring between 70k and 120k iterations. Employing the linear model 11 in PA helps to avoid such long lasting plateaus. The blue curves (with 11 in PA) exhibit smooth transitions when

---

[5]In fact, this architecture was initially selected by Chen et al. (2016) for MNIST ($28 \times 28$). In the released code of (Lučić et al., 2018), such InfoGAN architecture was adopted not only for MNIST and Fashion-MNIST, but also CIFAR10 and CELEBA with higher resolutions.

[6]We follow (Lučić et al., 2018), which excludes outliers while reporting the FID values.

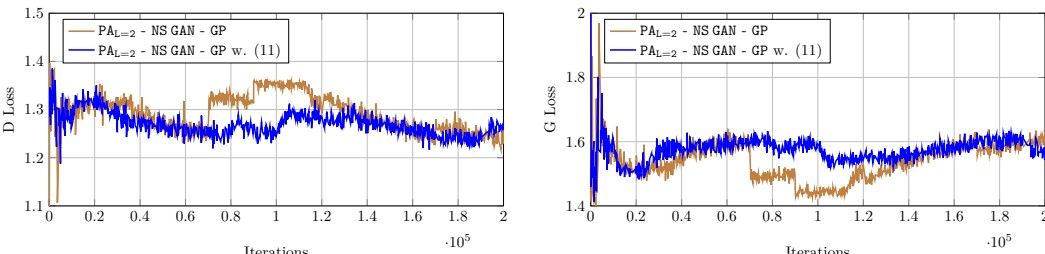

**Figure A10:** The behaviour of `D Loss` and `G Loss` over iterations on CIFAR10 using SN-DCGAN, with and without the linear model 11 in progression scheduling.

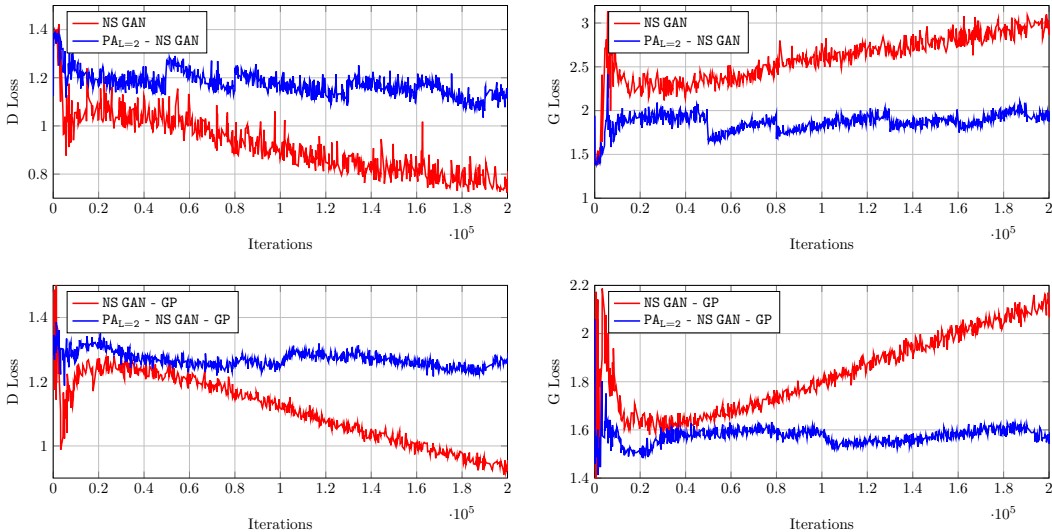

**Figure A11:** The behaviour of `D Loss` and `G Loss` of `NS GAN` and `NS GAN - GP` with and without PA during the training on CIFAR10 using the SN-DCGAN architecture.

the augmentation level increases. Overall, the linear model 11 in PA helps to speed up the learning process. The main motivation of having it is to take precautions against potential ill adaptation to new augmentation levels.

### A.7 EFFECT OF PA ON THE TRAINING STABILITY

Figure A11 compares `D Loss` and `G Loss` of `NS GAN` and `NS GAN - GP` with and without PA during the training on CIFAR10 using the SN-DCGAN architecture. Without PA the training becomes unstable over time independent of using GP (red curves), while employing PA helps to maintain a healthy competition between the discriminator and the generator. Observing the behaviour of `D Loss` and `G Loss`, we conclude that PA can effectively prevent the discriminator from overfitting, alleviating the vanishing gradient issue and thus enabling continuous learning of the generator.

In Table A11 we present mean and standard deviation values of FID across 50 independent runs on four datasets. We follow the work of (Lučić et al., 2018) reporting the numbers while excluding significant outlier runs. Excluding outlier runs mostly influences the standard deviation of the FID values. Table A11 shows that in the same setting as (Lučić et al., 2018) PA not only improves the FID values, but also reduces their standard deviations across multiple runs. This highlights that the training becomes more stable with PA. The improvement of using PA is consistent across four datasets, although the value of the mean FID heavily fluctuates dependent on the dataset.

### A.8 ANALYSIS OF THE TASK DIFFICULTY OF THE DISCRIMINATOR WITH PA

With PA we have casted the discrimination task into a checksum operation, involving two steps. The data and synthetic samples are combined with the augmentation bit sequence, resulting in data and

|  | Outliers removed | MNIST | Fashion-MNIST | CIFAR10 | CELEBA |
|---|---|---|---|---|---|
| NS GAN (Goodfellow et al., 2014) | ✓ | $6.8 \pm 0.5$ | $26.5 \pm 1.6$ | $58.5 \pm 1.9$ | $55.0 \pm 3.3$ |
| PA - NS GAN | ✓ | $8.4 \pm 0.6$ | $18.1 \pm 1.1$ | $44.3 \pm 1.3$ | $\mathbf{45.4 \pm 2.1}$ |
| PA - NS GAN (*) | ✓ | $\mathbf{6.5 \pm 0.4}$ | $\mathbf{15.8 \pm 0.7}$ | $\mathbf{42.8 \pm 1.3}$ | $\mathbf{45.4 \pm 2.1}$ |
| PA - NS GAN | ✗ | $8.8 \pm 1.1$ | $18.4 \pm 1.5$ | $44.6 \pm 1.9$ | $46.9 \pm 3.3$ |
| PA - NS GAN (*) | ✗ | $6.6 \pm 0.8$ | $15.8 \pm 1.1$ | $43.1 \pm 1.6$ | $46.8 \pm 3.2$ |

**Table A11:** Mean and standard deviation of FID values before and after excluding $15\%$ of outliers across 50 independent runs. For the four datasets (from left to right), the results are attained after 20, 20, 100 and 40 epochs, respectively, except for the PA results marked with (*). For (*) the training time is not constrained by the previously specified number of epochs, see A.5 for details.

synthetic samples contained in both true and fake classes. So, first the discriminator needs to recover the original sample and the augmentation bit sequence, and to decide if the original sample is the data or synthetic sample (this task is essentially identical to the original discrimination task). Second, the discriminator needs to learn that the recovered sample and the augmentation bit sequence jointly follow the checksum principle.

The difficulties of these two problems strongly depend on the way of feeding the augmentation bit sequence into the discriminator network. Providing the augmentation bit sequence to the input layer or lower layers generally makes the checksum operation more difficult than providing it to the upper layers. As the former has more difficulties decoupling the two tasks and trying to solve them jointly, while the latter can solve them sequentially. One naive design is to combine the bit sequence with the original output of the discriminator. In this case the checksum operation becomes nearly trivial and does not have a major influence on the training. The other way is to concatenate the augmentation bit sequence directly with the data and synthetic samples, ensuring that the task remains non-trivial for the discriminator.

**Toy Example: Binary Classification with PA.** For an illustration purpose, here we design a simple experiment. We employ discriminator for the classification task, omitting the use of the generator. Namely, two classes of CIFAR10 (cats and dogs) are extracted, conveying the bit 0 and 1. Together with the augmentation bit sequences, they are provided as input to the discriminator, which task is to perform binary classification. Using the SN-DCGAN architecture, we directly concatenate the augmentation bit sequences with the cat and dog images.

Figure A12 depicts the variation of the discriminator loss (`D Loss`) over the periodic progressive augmentation (every 2k iterations). It is evident that whenever a new augmentation level is reached an abrupt increase of `D Loss` is observed, showing that the task of the discriminator becomes harder with each level of PA.

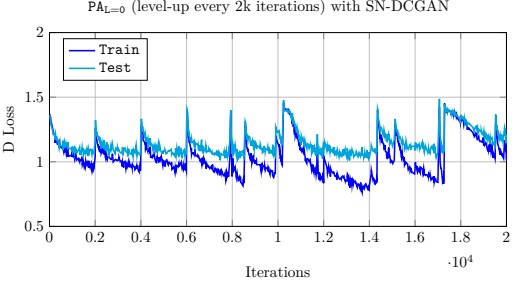 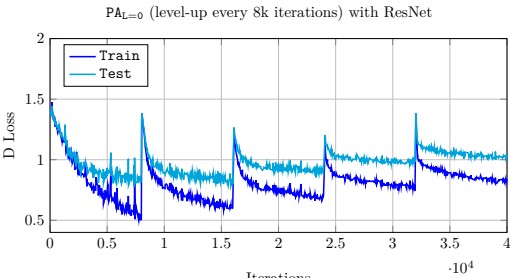

**Figure A12:** Binary classification on CIFAR10 with the presence of PA, using the SN-DCGAN architecture. The augmentation level is increased every 2k iterations.

**Figure A13:** Binary classification on CIFAR10 with the presence of PA, using the ResNet architecture (Kurach et al., 2018). The augmentation level is increased every 8k iterations.

**Employing Residual/Skip Connections with PA.** With the presence of skip connections, the augmentation bit sequence that is concatenated with the inputs will also be directly wired to the

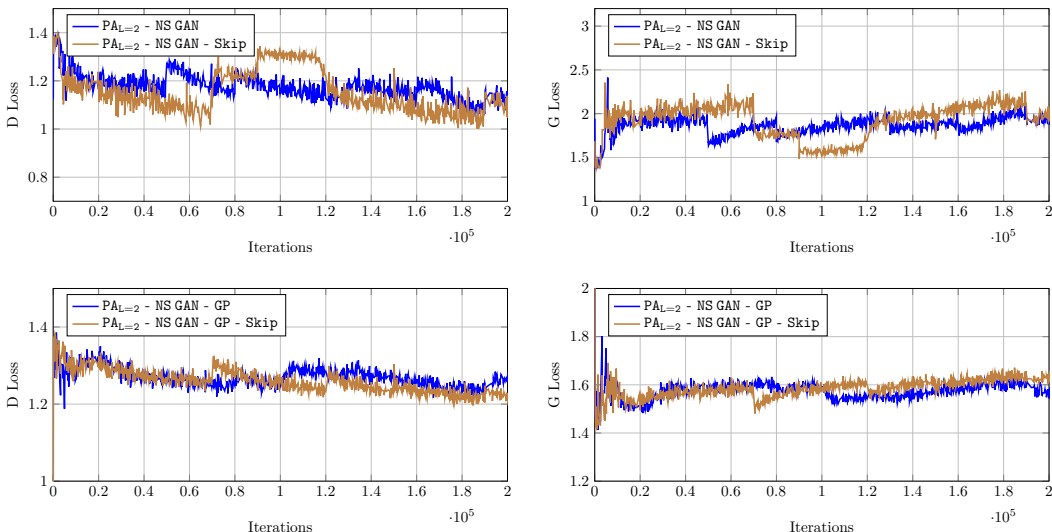

**Figure A14:** The behaviour of `D Loss` and `G Loss` of $PA_{L=2}$ - `NS GAN` and $PA_{L=2}$ - `NS GAN - GP` during the training on CIFAR10 using the SN-DCGAN architecture with and without skip connections.

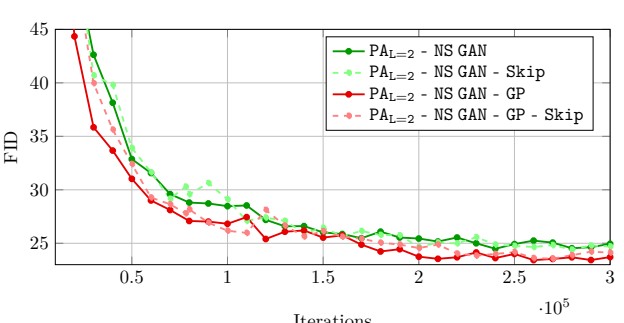

**Figure A15:** Mean FID values attained over iterations across five independent runs on CIFAR10 using the SN-DCGAN architecture with and without skip connections.

|  | 200k | 300k |
|---|---|---|
| `NS GAN` (Kurach et al., 2018) | 26.7 | — |
| `NS GAN` (ours) | 26.3 | 25.7 |
| $PA_{L=2}$ - `NS GAN` | 24.5 | 23.8 |
| $PA_{L=2}$ - `NS GAN - Skip` | **24.1** | **23.7** |
| `NS GAN - GP` (Kurach et al., 2018) | 26.2 | — |
| $PA_{L=2}$ - `NS GAN - GP` | **23.2** | **22.5** |
| $PA_{L=2}$ - `NS GAN - GP - Skip` | 23.8 | 23.0 |

**Table A12:** Median FID values attained on CIFAR10 using the SN-DCGAN architecture with and without skip connections.

network output. In the next two examples, we examine the task difficulty under such situation. First, we switch to using ResNet CIFAR10 (Kurach et al., 2018) for the toy example of binary classification of cats and dogs, while concatenating the augmentation bits with the inputs of the last residual block. By doing so, the skip connection will directly deliver this information to the network final layer. As we can see from Figure A13, the task remains difficult and PA persistently increases the task difficulty.

Next, we switch to the GAN setting and repeat the experiment of SN-DCGAN CIFAR10 in Sec. 5.1. Instead of only concatenating the augmentation bits with the input images, we also insert a skip connection, additionally concatenating them with the input to the last convolutional layer.[7] Depicting the `D Loss` and `G Loss` during the course of training, Figure A14 confirms that PA remains effective with the skip connection. As a result, a similar performance is reported in Figure A15 and Table A12. This showcases that the proposed PA generalizes well across different GAN design choices (including networks with residual or skip connections).

### A.9 COMPARISON WITH THE DROPOUT REGULARIZATION

In this section we compare the proposed PA with another regularization technique which also aims to weaken the discriminator, such as adding dropout at the first layer of the discriminator. Our

---

[7]XOR operation is a non-linear function. Therefore, we do not take the last linear layer.

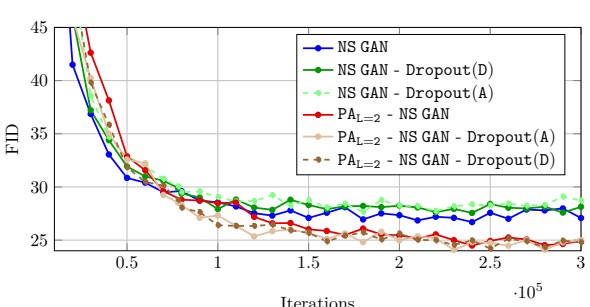

| | 200k | 300k |
|---|---|---|
| NS GAN (Kurach et al., 2018) | 26.7 | — |
| NS GAN (ours) | 26.3 | 25.7 |
| NS GAN - GP (Kurach et al., 2018) | 26.2 | — |
| NS GAN - Dropout(D) | 27.0 | 26.8 |
| NS GAN - Dropout(A) | 27.4 | 27.0 |
| $PA_{L=2}$ - NS GAN | 24.5 | 23.8 |
| $PA_{L=2}$ - NS GAN - Dropout(D) | 24.6 | 23.7 |
| $PA_{L=2}$ - NS GAN - Dropout(A) | 24.4 | 23.6 |

**Figure A16:** Mean FID values attained over iterations across five independent runs on CIFAR10 using SN-DCGAN with and without dropout.

**Table A13:** Median FID values attained on CIFAR10 using SN-DCGAN with and without dropout.

proposed augmentation scheme is orthogonal to it and thus can be applied along with the dropout regularization.

We adopt dropout at the output of the first convolutional layer of SN-DCGAN and experiment with two configurations. In the first configuration the dropout rate is linearly increased from zero to $0.7$, we call it an ascending mode (A). In the second the dropout rate starts from $0.7$ and linearly descends to zero during the training, we call it a descending mode (D). From Figure A16 and Table A13 we observe that neither of the two dropout modes improves the performance of NS GAN, whereas PA shows to be more effective and preserves the performance gain even in combination with the dropout.

