# OpenReview forum: "PA-GAN: Improving GAN Training by Progressive Augmentation"
_ICLR.cc/2019/Conference_

### Official Review · AnonReviewer2 · 2018-10-29
**Provide an additional bitstring to the discriminator which can swap the label for observed and generated samples.**

**Rating:** 5
**Confidence:** 4

**Review:**

Authors argue that the main issue with stability in GANs is due to the discriminator becoming too powerful too quickly. To address this issue they propose to make the task progressively more difficult: Instead of providing only the samples to the discriminator, an additional (processed) bitstring is provided. The idea is that the bitstring in combination with the sample determines whether the sample should be considered true or fake. This in turn requires the decision boundary of the discriminator to become more complicated for increasing lengths of the bitstring. In a limited set of experiments the authors show that the proposed approach can improve the FID scores.

Pro:
- A simple idea to make the problem progressively more difficult.
- The writing is relatively easy to follow.
- Standardized experimental setup.

Con:
- Ablation study of the training tricks is missing: (1) How does the proposed approach perform when no progressive scheduling is used? (2) How does it perform without the linear model for increasing p? (3) How does the learning rate of G impact the quality? Does one need all of these tricks? Arguably, if one includes the FID/KID to modify the learning rates in the competing approaches, one could find a good setup which yields improved results. This is my major issue with this approach.
- Clarity can be improved: several pages of theory can really be summarized into “learning the joint distribution implies that the marginals are also correctly learned’ (similar to ALI/BIGAN). This would leave much more space to perform necessary ablation studies.
- Comparison to [1] is missing: In that model, it seems that the same effect can be achieved and strongly improves the FID. Namely, they introduce a model in which observed samples pass through a "lens" before being revealed to the discriminator thus balancing the generator and discriminator by gradually revealing more detailed features.
- Can you provide more convincing arguments that the strength of the discriminator is a major factor we should be fixing? In some approaches such as Wasserstein GAN, we should train the discriminator to optimality in each round. Why is the proposed approach more practical then approaches such as [2]?

[1] http://proceedings.mlr.press/v80/sajjadi18a.html
[2] https://arxiv.org/abs/1706.08500

---

> ### Author Response · Authors · 2018-11-25
> **Response to R2**
>
> Apart from the questions that have been addressed in our general response (A-C), here are answers to R2's remaining questions:
>
> 1) Several pages of theory can really be summarized into “learning the joint distribution implies that the marginals are also correctly learned’ (similar to ALI/BIGAN).
>
> From the perspective of joint distribution matching, ALI/BiGAN constructs two joint distributions which marginals are with respect to the data and model distribution. Therefore, when two joint distributions are mutually matched, the model distribution approaches the data distribution. In our case, as mathematically shown in Appendix A.1, the two joint distributions have identical marginals by construction. Upon a completely different line, we prove its equivalence to the original problem of GAN.
> Furthermore, ALI/BIGAN aim at generative latent modeling, whereas we aim at stabilizing the training process of GAN (generative modeling without the latent code). The augmentation bit sequence s in our case is not a latent code of the input image. The generator does not take s as its input to generate the synthetic data. A successful training of ALI/BIGAN requires x and z being mutually dependent, e.g., its variant ALICE enforcing the dependence through conditional entropy. We are in the opposite situation, namely, x and s being mutually independent implies a perfect generator.
>
> 2) “Can you provide more convincing arguments that the strength of the discriminator is a major factor we should be fixing?  In some approaches such as Wasserstein GAN, we should train the discriminator to optimality in each round.”
>
> We first would like refer the reviewer to works of [Arjovsky & Bottou ICLR’17 and Sønderby et al. ICLR’17]. In these works, the authors provide the explanation of the fundamental problem of instability of GANs and explain why additional techniques are required to weaken the discriminator. In particular, the problem is that the support of the real data distribution and the generative model distribution are often non-overlapping (in image modelling, distribution of natural images is often assumed to be concentrated on or around a lower-dimensional manifold). In such situations, the divergences which GANs are minimizing become meaningless (the Jensen-Shannon divergence is saturated so its maximum value), and the discriminator is extremely prone to overfitting, which can lead to instabilities. One of the ways to avoid this behavior is to weaken the discriminator by making its job harder, which we successfully achieve by performing PA.  Recent work address the problem of weakening the discriminator in two ways, which are orthogonal to our approach, either by regularizing the discriminator via different variations of the gradient penalty [Gulrajani et al. 2017, Roth et al. 2017, Fedus et al. 2018] or altering directly the data samples [Arjovsky & Bottou 2017, Sønderby et al. 2017, Sajjadi et al. 2018].
>
> The Wasserstein distance belongs to the family of integral probability metrics, which are well defined in contrast to f-divergences [Arjovsky & Bottou 2017]. Thus, [Arjovsky et al. 2017] advises to train the discriminator to optimality. However, in WGAN the class of discriminators is restricted to Lipschitz continuous functions, which yields a hard constraint on the function class that is empirically hard to satisfy. Moreover, [Mescheder et al. 2018] show that WGANs (and WGAN-GPs) do not converge as in practice the discriminator is trained with a fixed number of discriminator updates per generator update and thus the discriminator optimality is not guaranteed. Therefore, preserving a healthy competition between the generator and discriminator remains challenging (the escalation of signal magnitudes in the generator and discriminator is still observed in practice [Karras et al. 2018]). PA can serve as a regularization technique in this case. Empirically we show that employing PA with WGAN and WGAN-GP leads to better performance.
>
> 3) Why is the proposed approach more practical then approaches such as https://arxiv.org/abs/1706.08500
>
> Different learning rates of the generator and discriminator (or different number of updates) introduce an additional hyper-parameter which requires a careful tuning to achieve a performance improvement. In contrast, our progressive augmentation is done automatically by a simple threshold test of the KID score. We would like to point out, that it is possible to combine both approaches, which potentially might be benefiting to each other. We consider the comparison/combination with the two time-scale update rule as part of the future work.
>
> We thank R2 for the comments and suggestions that allowed us to greatly improve the quality of the work.

---

### Official Review · AnonReviewer1 · 2018-10-30
**Some major flaws in the approach**

**Rating:** 5
**Confidence:** 5

**Review:**

This paper proposes a new trick to improve the stability of GANs. In particular the authors try to tackle the vanishing gradient problem in GANs, when the discriminator becomes to strong and is able to perfectly separate the distribution early in training, resulting in almost zero gradient for the generator. The authors propose to increase the difficulty of the task during training to avoid the discriminator to become too strong.

The paper is quite well written and clear. However there is several unsupported claims (see below).

A lot of work has been proposed to regularize the discriminator, it's not clear how different this approach is to adding noise to the input or adding dropout to the discriminator.

Pros:
- The experimental section is quite thorough and the results seem overall good.
- The paper is quite clear.

Cons:
- There is a major mistake in the derivation of the proposed method. In eq. (6) & (7), (c) is not an equivalence, minimizing the KL divergence is not the same as minimizing the Jensen-Shannon divergence. The only thing we have is that: KL(p||q) = 0 <=> JSD(p||q) = 0 <=> p=q . The same kind of mistake is made for (d). Note that the KL-divergence can also be approximated with a GAN see [1]. Since the equivalence between (6) and (7) doesn't hold, the equation (11) doesn't hold either.

- The authors say that the discriminator can detect the class of a sample by using checksum, the checksum is quite easy for a neural networks to learn so I don't really see how the method proposed actually increase the difficulty of the task for the discriminator. Especially if the last layer of the discriminator learns to perform a checksum, and the discriminator architecture has residual connections, then it should be straight-forward for the discriminator to solve the new task given it can already solve the previous task. So I'm not sure the method would still works if we use ResNet architecture for the discriminator.

- I believe the approach is really similar to adding noise to the input. I think the method should be compared to this kind of baseline. Indeed the method seems almost equivalent to resetting some of the weights of the first layer of the discriminator when the discriminator becomes too strong, so I think it should also be compared to other regularization such as dropout noise on the discriminator.

- The authors claim that their method doesn't "just memorize the true data distribution". It's not clear to me why this should be the case and this is neither shown theoretically or empirically. I encourage the author to think about some way to support this claim.

- The authors states that "adding high-dimensional noise introduces significant variance in the parameter estimation, which slows down training", can the author give some references to support that statement ?

- According to the author: "Regularizing the discriminator with the gradient penalty depends on the model distribution, which changes during training and thus results in increased runtime". While I agree that computing the gradient penalty slightly increase the runtime because we need to compute some second order derivatives, I don't see how these increase of runtime is due to change in the model distribution. The authors should clarify what they mean.

Others:
- It would be very interesting to study when does the level number increase and what happens when it increase ? Also what is the final number of level at the end of training ?

Conclusion:
The idea has some major flaws that need to be fixed. I believe the idea has similar effect to adding dropout on the first layer of the discriminator. I don't think the paper should be accepted unless those major concerns are resolved.

References:
[1] Nowozin, S., Cseke, B., & Tomioka, R. (2016). f-gan: Training generative neural samplers using variational divergence minimization. NIPS

---

> ### Author Response · Authors · 2018-11-25
> **Response to R1**
>
> Apart from the questions that have been addressed in our general response (A-C), here are answers to remaining questions of R1:
>
> 1) "adding high-dimensional noise introduces significant variance in the parameter estimation, which slows down training", can the author give some references to support that statement?
>
> We would like to refer R1 to [Roth et al. NIPS’17]. This work argues that “high-dimensional noise introduces significant variance in the parameter estimation process“, “counteracting this requires a lot of samples and therefore ultimately leads to a costly or impractical solution“  and that “explicitly adding noise in high-dimensional ambient spaces introduces additional sampling variance”, which leads to the increase of the overall training time.
>
> 2) According to the author: "Regularizing the discriminator with the gradient penalty depends on the model distribution, which changes during training and thus results in increased runtime". I don't see how these increase of runtime is due to change in the model distribution. The authors should clarify what they mean.
>
> Here we meant that there are two drawbacks of employing the gradient penalty [Kurach et al. 2018]. First, it can depend on the model distribution, which changes during training. Second, computing the gradient norms results in increased runtime.
>
> 3) The authors claim that their method doesn't "just memorize the true data distribution". It's not clear to me why this should be the case and this is neither shown theoretically or empirically. I encourage the author to think about some way to support this claim.
>
> Our aim was to provide the intuition to the reader why we think employing PA result in better FID and inception scores, which are known to correspond to the variation of generated images. We believe that structurally augmenting the input sample space and mapping it to higher dimensions encourages the generator to explore various paths towards the true data distribution, leading to the improved variation of generated images, which we observe in the improved FID and IS metrics.
>
> We thank R1 for pointing out the confusing statements mentioned above, we adjusted those statements in the in the revision to improve the overall clarity.

---

> > ### Comment · AnonReviewer1 · 2018-12-05
> > **reply**
> >
> > The proof in the appendix is correct and I thus raised my score, however I think there is still some points which remain unclear to me and I think need some clarifications:
> >
> > 1. I think the theory proposed is misleading about why the method works. Indeed as mentioned by myself and the other reviewers, the additional bit can easily be learnt if passed to the last layer instead of the first layer. Thus I believe what's really important here is that you feed this additional bit to the first layer, and this is not explained by the current theory you proposed. Thus I believe there is something missing in your explanation.
> >
> > My intuition is that actually the method proposed is really close to other regularization techniques but does it in a "smarter" way. If you consider that the first layer is a convolutional layer then the output is a sum over all the channels of the input multiplied by the corresponding filter. Adding one channel correspond to adding one element in that sum. This new input channel is either 0 or 1, thus either we add this new channel or we don't add it with some probability p(s). This seems really similar to dropout but instead of resetting the weights to 0, this approach resets the weights to a new value that is also learned. Thus it's like having a mixture over the parameters of the first layer, where we add a mixture component during training according to some "smart" rule.
> > My question is thus the following what would happen if you were to increase the input dimension as done in the paper but keep the classification problem for the discriminator the same (basically adding this new channel for all inputs with some probability p and just classifying between Pd(x) adn Pg(x))?
> >
> > 2. "encourages the generator to explore various paths towards the data distribution" this claim is never shown either theoretically or experimentally, I think it should be removed.

---

> > > ### Author Response · Authors · 2018-12-06
> > > **Respond to R1's further comments on the revision**
> > >
> > > First of all, we thank R1 for acknowledging the revision of the paper and correctness of the proof. In the following, we answer further questions and concerns of R1.
> > >
> > > 1.
> > >
> > > - The theory presented in the paper is itself agnostic to the network architecture and means to show that with the progressive augmentation of the input space the original objective of the discriminator is preserved. We would like to highlight that in our approach we proposed to augment the input space only. As illustrated in Fig.1, we can clearly observe from the toy example that the discrimination task becomes more challenging in this case as decision boundary becomes more complicated. In the appendix, we further show experimentally that the task of the discriminator becomes gradually harder with each augmentation level. We also explored in the appendix alternative ways of providing the augmentation bits into the network, confirming the effectiveness of our proposed approach. We agree with R1 that it is worth further investigating PA across different network designs, in particular optimizing it towards a specific application. However, we consider this a future work.
> > >
> > > -  The intuition of R1 is correct in terms of how the first layer tries to process the augmentation bits. However, if the task of the discriminator is to only classify x (the classification problem for the discriminator is kept the same), it can easily learn to set the weights for the augmentation channels to zeros, as they are irrelevant redundant information for making a classification decision. In other words, after few initial iterations the training boils down to the classic GAN case. We ran experiments in the CIFAR10 INFOGAN case. As expected, the achieved FID score 57.1 \pm 2 is on par with the original NS-GAN FID score 58.5\pm 1.9, while the proposed PA improves it to 44.6 \pm 1.9 as reported in Table 2.
> > >
> > > - Adding or multiplying some feature maps with some random variables are non-invertible distortions to the learning process. With the target of classifying x only, the discriminator is trained to be invariant to them, thereby avoiding over-fitting. However, the augmentation bits do not affect the learning process of the discriminator as non-invertible distortions. Given the way they are fed into the network, the discriminator can easily ignore them by setting the associated weights to zeros. The key is: They carry information that influence the classification decision. This makes the proposed method different to stochastic regularization techniques, which introduce random variables to the network and remain the task of classifying x only.
> > >
> > > - We believe our method does not even try to “regularize” the discriminator, as the augmentation bits do not aim to smoothen the decision boundary specified by the discriminator. Instead, they make the decision boundary lie in a higher dimensional space. Both GP and spectral normalization are known to be effective regularization techniques for GANs and we experimentally show the complementary of the proposed PA to them. It is worth noting that on the higher resolution dataset CelebaHQ (128x128) the gain of employing PA is much larger than for CIFAR10. This shows a good potential of this approach for more challenging applications.
> > >
> > > 2.
> > > We agree that it would be beneficial to illustrate this point experimentally as well. Intuitively, for each realization of the augmentation bit the generator sees a different loss function to minimize, thereby providing it diverse paths to approach the data distribution. Since the augmentation bits affect the classification process of the discriminator, this plays a role of ensuring the diversity in the paths. This is different to the cases where the discriminator is trained to be invariant to the random variables involved in the network. Upon revision, we will remove this claim to avoid any misunderstandings.

---

### Official Review · AnonReviewer3 · 2018-11-04
**Interesting idea, but lacking theoretical support or sufficient empirical analysis.**

**Rating:** 4
**Confidence:** 2

**Review:**

This paper modifies the GAN objective by defining the TRUE and FAKE labels in terms of both the training sample, and a newly introduced random variable s. The intuition is that by progressively changing the definition of s, and its effect on the label, we can prevent the discriminator network from immediately learning to separate the two classes.

The paper doesn't give any strong theoretical support for this intuition. And it I found it a bit surprising that the discriminator doesn't immediately learn the one extra bit of information introduced by every new level of augmentation. However, the results do seem to show that this augmentation has a beneficial effect on two different architectures in different data scenarios, although the increase is not uniform over all settings.

The approach presented in this paper is motivated primarily as a method of increasing stability of training but this is not directly investigated. Figure 3 and Table 2 both suggest that the augmentation does nothing to reduce variance between runs. There is also no direct comparison to other methods of weakening the discriminator, although these are mentioned in the related work. I think the paper would be much improved by a thorough investigation of the method's effect on training stability, to go along with the current set of evaluations.

---

> ### Author Response · Authors · 2018-11-25
> **Reponse to R3**
>
> We thank R3 for the reviewing effort and comments that allowed us to improve the quality of our work. We have addressed R3's questions in our general response (A-C).

---

### Author Response · Authors · 2018-11-25
**General Response**

We thank reviewers for their feedback indicating that the paper is well written (R1) and easy to follow (R2), with thorough experiments (R1) in the standardized setup (R2); that the proposed approach is interesting (R3) and simple (R2), but yet makes the task of discriminator progressively more difficult (R2) and hence has a beneficial effect on GAN training across different architectures in different data scenarios (R3), leading to overall good results (R1).

Despite the positive comments, we believe some important points of the paper have been missed or misunderstood by the reviewers.  Thus, we would like to re-state our contributions and invite the area chair and the reviewers to re-assess our work. Our contributions are:
-	We propose a novel method - progressive augmentation of GANs (PA-GAN), which addresses the important problem of maintaining a healthy competition between the generator and discriminator, leading to a better quality generator.
-	We experimentally show the effectiveness of the PA-GAN and report the pronounced performance improvement across different datasets (~20 % on average, see Table 2 ) using the standardized setup (performing multiple independent runs while avoiding additional hyper-parameter tuning).
-	We theoretically prove that the proposed approach preserves the original GAN objective (see Sec. 4.1 and Appendix A.1) and in contrast to other techniques  does not bias the optimality of the discriminator by modifying the real data samples or introducing noise to the input (such as [Sajjadi et al. ICML’18] or [Arjovsky & Bottou ICLR’17 and Sønderby et al. ICLR’17]).
-	The proposed technique can be easily integrated into different GAN frameworks (including networks with residual or skip connections) with minimal changes (see Sec. 5 and Appendix A.8).
-	Our technique is orthogonal to existing work, it can be successfully employed with other regularizations strategies, e.g., gradient penalty and spectral normalization, which we experimentally show in Sec. 5.

---

> ### Author Response · Authors · 2018-11-25
> **High-level Comments (Part A)**
>
> Next, we address some of the high-level concerns raised and requested clarifications.
>
> 1)	R1: “a major mistake in the derivation”:
>
> R1 is concerned with the validity of the proof in Sec. 4.1 and overall derivation of the method, questioning the equivalence in eq. (6). We argue that the provided equivalence is valid and believe that the confusion comes from the shortening of the proof due to the page limit. To improve the clarity, we altered the explanation in sec. 4.1 and provided a detailed derivation of eq. (6) in Appendix A.1. We invite R1 to re-asses its quality and hope the detailed explanation will clarify the original confusion.
>
> Below we briefly summarize the derivation presented in the paper:
> Introducing a random bit s, we have two ways to associate s=0 and s=1 with the sample x~Pd and x~Pg, respectively. These two ways simply define two joint distributions P(x, s) and Q(x, s). The JS divergence between Pd and Pg is equal to the JS divergence between P(x,s) and Q(x,s), for any given pair of (Pd and Pg). Therefore, it is equivalent to minimize either of the two JS divergences for optimizing G.
> To show that the two JS divergences are identical, we based on 1) the connection between JS divergence and mutual information, 2) mutual information is a KL divergence, 3) 0.5 P(x, s) + 0.5 Q(x, s) = Pm(x) P(s), where Pm(x) is the marginal of P(x, s) and Q(x, s). It is important to note that the marginal of P(x,s) and Q(x,s) with respect to x are always identical by construction and matching P(x,s) and Q(x,s) does not fall into the framework of ALI/BiGAN.
>
> 2) R1, R3:”surprising why D doesn’t immediately learn the new task”, “how the method proposed actually increase the difficulty of the task for the discriminator?”
>
> Although the naive checksum (XOR) operation between two bits is an easy task for the neural network, in case of the PA-GAN setting additional challenges arise.  First, the task of the discriminator D becomes two folded: it not only needs to decide if the image is real or fake, but on top of it to figure out a new task – perform a checksum operation between the image input x and random bit s.  Second, to perform the checksum D also needs to learn how to separate the input into x and s, making the task more difficult (even for networks with residual and skip connections, see Appendix A.8).
>
> The way we provide the random bit s to the discriminator plays the role how fast it is able to learn the task. In our proposed implementation s is concatenated with the data point x and provided to the discriminator D as an input, making harder for D to make separation between x and s. We also experimented with providing s only to the intermediate layers or just to the last layer of D. Our observations are aligned with the intuition of R1: providing s directly to the last layer makes the task easy for D, as it needs now only to solve the checksum between two bits (the task for x has already been solved and separation between x and s is trivial) and does not result in any improvement.
> In addition, to figure out the task D needs to see at least 4 times more data points (all input combinations for XOR operation).  We can leverage this to control the task difficulty by regulating the balance between classes in the batch (by biasing uniform sampling of s).
>
> At last, we also would like to point out that we do not claim that performing checksum operation is a difficult task for the discriminator. We only claim that with each level of the augmentation the task becomes gradually harder and it takes some time for the discriminator to reach the decision confidence. When the discriminator learns the task the next augmentation step is performed. This strategy allows to maintain the healthy balance between the generator and discriminator. We show experimentally that it is beneficial to use PA-GAN.
>
> We provide the detailed analysis of the task difficulty of the discriminator with PA in Appendix A.8, where we experimentally show that that the task of the discriminator becomes harder with each level of PA.

---

> > ### Author Response · Authors · 2018-11-25
> > **High-level Comments (Part B)**
> >
> > 3) R1: “if the discriminator architecture has residual connections, then it should be straight-forward for the discriminator to solve the new task given it can already solve the previous task”
> >
> > In Appendix A.8 we provide experiments using the discriminator networks with and without residual connections while performing the proposed progressive augmentation.  We show that the task remains difficult and it is not straightforward for the discriminator with residual connections to learn the task (see Fig. A12 and A13). Even though the checksum operation might become easier with residual connections, the separation of the input into x and s still remains challenging for the discriminator.
> >
> > Additionally, we also experimented with adding skip connections to the discriminator. However, we did not observe any degradation in the performance of PA-GAN (see Fig. A14-A15 and Table A12). This showcases that the proposed PA generalizes well across different architectures, supporting our claims in the paper. We refer R1 to Appendix A.8 for more details.
> >
> > 4) R1, R2, R3: Difference to other methods of weakening the discriminator:
> >
> > -  R1: “how different this approach is to adding noise to the input or adding dropout to the discriminator”, R3: “no direct comparison to other methods of weakening the discriminator”
> >
> > Both adding noise and dropout are regularization techniques. In the non-overlapped space of Pd and Pg, they both attempt to let the discriminator make a soft decision. As such, the support of Pd will not be surrounded by a zero-gradient area, preventing Pg from approaching it. Our augmentation scheme is orthogonal to them. We propose to increase the problem dimensionality. Initially, the discriminator needs to separate two inputs living in an N-dimensional space. Now, the separation is conducted in an (N+L)-dimensional space, where L is the augmentation level. In other words, the intrinsic dimension of the discriminator function is increased, see Figure 1. Since we do not regularize the discriminator, the decision boundary can still be very sharp. Therefore, PA can be applied along with the regularization techniques.
> >
> > In Sec. 5, we have experimented on using PA with gradient penalty (GP), which is shown to be an effective regularization technique for the discriminator [Kurach et al. 2018]. The achieved gain is more pronounced than using either PA or GP alone. This empirically confirms that PA and regularization techniques are orthogonal and more importantly mutually beneficial.
> >
> > As discussed by Roth et al. NIPS2017, adding high-dimensional noise can introduce significant variance in the parameter estimation process. By analytically convolving the Gaussian noise distribution with the data and generative model distributions, they alternatively proposed a regularizer, adding it to the discriminator loss function together with an annealing process to adjust the weighting factor. The regularizer resembles GP that has been successfully combined with PA in Sec. 5, e.g., see Figure 3. One difference to GP is that it is zero-centered.
> >
> > - R2: “difference to [Sajjadi et al. ICML’18]”
> > Sajjadi et al. proposed to blur the input and gradually remove the blurring effect during the course of training. We believe that this technique has some common ground with employing instance noise, as both techniques perform direct modifications on the data samples. In contrast, PA does not directly modify the data samples, but rather structurally appended to them increasing the input dimensionality of the discriminator. As we have mentioned previously, our approach is orthogonal to the techniques that directly modify the data samples and in principle can be combined with them.
> >
> > In this revision, we additionally provide new results of combining PA with dropout in Appendix A.9. Figure 16 shows that dropout alone does not improve the performance of NS-GAN, in fact, causing a small performance degradation. Whereas PA shows to be more effective and preserves the performance gain even in combination with the dropout. Analogous to our early observations with GP, this shows that PA can work in combination with different regularization techniques.
> > The combination of the zero-centered GP of Roth et al. NIPS2017  with PA requires further investigation. Unlike the GAN case, the two distributions to be matched are not simply the data and generative model distributions. In order to properly take into account the augmentation bits, we need to revise the derivation of the zero-centered GP.

---

> > > ### Author Response · Authors · 2018-11-25
> > > **High-level Comments (Part C)**
> > >
> > >
> > > 5) Ablation studies
> > >
> > > - R1: It would be very interesting to study when does the level number increase and what happens when it increase ?what is the final number of level at the end of training
> > >
> > > In this work, we use the KID score as a metric to decide when to increase the augmentation level. As such, we can let the augmentation take place whenever there is a need.  In principle, other metrics for evaluating the performance of GAN are applicable as well, but they are not the focus of this work. As mentioned in the paper for the reported experiments in Table 1 and Table 2 the maximal augmentation level reached was 7, and for the new experiments on CELEBA-HQ it was 10.
> > >
> > > - R3: the paper would be much improved by a thorough investigation of the method's effect on training stability
> > >
> > > We try to address these questions in the new section A.7 in the Appendix. In Figure A11, we analyze the loss of the discriminator and the generator over iterations with and without PA. Without PA, we can clearly observe that the loss of the discriminator quickly reduces after initial several thousand iterations. Accordingly, the generator loss increases. Such observation takes place even if we have adopted spectral normalization in the discriminator network.  On the other hand, our proposed PA can effectively prevent the discriminator from over-fitting. Whenever a new augmentation level is reached, the discriminator loss increases again while the generator loss reduces. Overall, through progression, we can on-the-fly control the two-player game between the generator and the discriminator.
> > >
> > > 6) R2: (1) How does the proposed approach perform when no progressive scheduling is used? (2) How does it perform without the linear model for increasing p?  (3) How does the learning rate of G impact the quality? Arguably, if one includes the FID/KID to modify the learning rates in the competing approaches, one could find a good setup which yields improved results.
> > >
> > > We address the first two questions in ablation studies in Appendix A.6:
> > > In Fig. A8, we empirically illustrate the importance of progressively increasing the augmentation levels to maintain the balance between the discriminator and the generator. In this example, the initial augmentation only helps to improve the performance at early iterations, however, the performance starts degrading once the discriminator learns the task but no progression is scheduled.
> > > In Fig. A10, we depict the loss of the discriminator over training iterations. Without the linear model for increasing p, we observe the discriminator can require a considerable number of iterations to pick up the new augmentation bit. Whenever this takes place, it can no longer guide the generator to learn the data distribution. The linear model helps to speed up the learning process, taking precautions against potential ill adaptation to new augmentation levels.
> > >
> > > Regarding the third question,  in [2], the authors adopted KID score to adapt the learning rate and reported KID=0.015 for CIFAR10 plus SNDCGAN. With our proposal, we are able to achieve KID=0.013. Since KID is not the primary evaluation metric for the quality of synthetic data, this number is only reported in the Appendix A.3 (see Table A5  and Figure A3).
> > > [2] M. Binkowski et. al. “Demystifying MMD GANs”, ICLR’2018.
> > >
> > > 7) R3: Figure 3 and Table 2 both suggest that the augmentation does nothing to reduce variance between runs.
> > >
> > > Here we would like to note that the numbers reported in Table 2 except our results with PA are the outcome of removing outliers among 50 independent runs (the numbers are taken from Lucic et al. 2018).  In Table A11, we did a similar analysis while removing the outliers.  Table A11 shows that in the same setting as Lucic et al. 2018 PA not only improves the FID values, but also reduces their standard deviations across multiple runs, highlighting that PA is helpful to improve the training stability.
> > >
> > > We additionally address individual concerns of reviewers by directly responding to the reviewers comment (see below).

---

### Public Comment · ~Daniel_Fojo1 · 2018-12-27
**Reproducibility Challenge**

We are a team of students from the Polytechnic University of Catalonia participating in the ICLR’19 reproducibility challenge (https://reproducibility-challenge.github.io/iclr_2019/). We are interested in reproducing the results of this paper.

We are happy to say that we could implement the main structure of PA-GAN, and observed an improvement with respect to a regular GAN. However, we could not achieve the same values for the FID and KID metrics yet. We have some questions about the paper:

In the paper, it is stated that it is necessary to decrease the Generator’s learning rate when achieving a new augmentation level. However, in the experiments section we could not find how this decrease should be done. We also wondered if a standard learning rate decay is used.

The following sentence is found in the paper:  “If the current KID score is less than 5% of the average of the two previous ones attained at the same augmentation level, the augmentation is levelled up, i.e. L is increased by one” We are not sure if this is right. For our implementation, we used the difference between the current KID and the average of the previous ones instead of the current one. Maybe it is needed to explain it in a more clarified way.

Two different learning rates are stated for the CIFAR10 experiment using SN-DCGAN (2e-4 and 4e-4).

You can find our implementation here https://github.com/telecombcn-dl/2018-dlai-team1 . Thank you.

---

> ### Author Response · Authors · 2018-12-30
> **Reply to Experiment Reproduction**
>
> Thanks for your interest in this paper.
>
> 1) Regarding your first question on adaptive learning rate: Since the two baseline works that we compared with used fixed learning rates for both the discriminator and the generator, we stick to such setup in our experiments as well for the sake of fair comparison. Certainly, our approach can work with adaptive learning rate. However, we did not claim that it is a "must" to adjust the learning rate during the course of progressive augmentation. It is advised, as it can be beneficial to speed up the adaptation to the new augmentation level, being analogous to the linear model (11) and as illustrated in Fig. A10.
>
> 2) You stated that "we used the difference between the current KID and the average of the previous ones instead of the current one". From your description, we believe that your understanding is correct. Maybe it is good to further note that the current KID and the previous two KID measures shall be attained at the same augmentation level.
>
> 3) For the SN-DCGAN experiment, the learning rate is 2e-4 for both the discriminator and generator. It is the best learning rate selected by Kurach et al., 2018.

---

> > ### Public Comment · ~Daniel_Fojo1 · 2018-12-31
> > **Reply to Experiment Reproduction**
> >
> > Thank you very much for your help.

---

### Meta-Review · Area_Chair1 · 2018-12-14

**Confidence:** 3
**Recommendation:** Reject

**Metareview:**

The submission hypothesizes that in typical GAN training the discriminator is too strong, too fast, and thus suggests a modification by which they gradually increases the task difficulty of the discriminator. This is done by introducing (effectively) a new random variable -- which has an effect on the label -- and which prevents the discriminator from solving its task too quickly.

There was a healthy amount of back-and-forth between the authors and the reviewers which allowed for a number of important clarifications to be made (esp. with regards to proofs, comparison with baselines, etc). My judgment of this paper is that it provides a neat way to overcome a particular difficulty of training GANs, but that there is a lot of confusion about the similarities (of lack thereof) with various potentially simpler alternatives such as input dropout, adding noise to the input etc. I was sometimes confused by the author response as well (they at once suggest that the proposed method reduces overfitting of the discriminator but also state that "We believe our method does not even try to “regularize” the discriminator"). Because of all this, the significance of this work is unclear and thus I do not recommend acceptance.